# Digital twin mathematical models suggest individualized hemorrhagic shock resuscitation strategies
Jeremy W. Cannon [1,2] ✉, Danielle S. Gruen [3,4], Ruben Zamora[3,4,5], Noah Brostoff[6], Kelly Hurst[6], John H. Harn[6], Fayten El-Dehaibi[3], Zhi Geng[1], Rami Namas[3,4], Jason L. Sperry[3,4], John B. Holcomb[7], Bryan A. Cotton[8], Jason J. Nam[9], Samantha Underwood [10], Martin A. Schreiber[10], Kevin K. Chung[11], Andriy I. Batchinsky [12], Leopoldo C. Cancio[13], Andrew J. Benjamin[14], Erin E. Fox[8], Steven C. Chang[6], Andrew P. Cap[9] & Yoram Vodovotz [3,4,5,15]

## Abstract

**Background** Optimizing resuscitation to reduce inflammation and organ dysfunction following human trauma-associated hemorrhagic shock is a major clinical hurdle. This is limited by the short duration of pre-clinical studies and the sparsity of early data in the clinical setting.

**Methods** We sought to bridge this gap by linking preclinical data in a porcine model with clinical data from patients from the Prospective, Observational, Multicenter, Major Trauma Transfusion (PROMMTT) study via a three-compartment ordinary differential equation model of inflammation and coagulation.

**Results** The mathematical model accurately predicts physiologic, inflammatory, and laboratory measures in both the porcine model and patients, as well as the outcome and time of death in the PROMMTT cohort. Model simulation suggests that resuscitation with plasma and red blood cells outperformed resuscitation with crystalloid or plasma alone, and that earlier plasma resuscitation reduced injury severity and increased survival time.

**Conclusions** This workflow may serve as a translational bridge from pre-clinical to clinical studies in trauma-associated hemorrhagic shock and other complex disease settings.

## Plain language summary

Research to improve survival in patients with severe bleeding after major trauma presents many challenges. Here, we created a computer model to simulate the effects of severe bleeding. We refined this model using data from existing animal studies to ensure our simulations were accurate. We also used patient data to further refine the simulations to accurately predict which patients would live and which would not. We studied the effects of different treatment protocols on these simulated patients and show that treatment with plasma (the fluid portion of blood that helps form blood clots) and red blood cells jointly, gave better results than treatment with intravenous fluid or plasma alone. Early treatment with plasma reduced injury severity and increased survival time. This modelling approach may improve our ability to evaluate new treatments for trauma-associated bleeding and other acute conditions.

---

Traumatic injury accompanied by hemorrhagic shock (T/HS) remains the leading cause of preventable death in both military and civilian trauma patients[1]. In these patients, death occurs early (within one hour)[2], and for early survivors hemorrhage and trauma induce an acute inflammatory response that ultimately drive multiple organ dysfunction and death much like sepsis. This response promotes a coordinated mobilization of numerous circulating mediators and inflammatory cells, precipitating a cascade of generally deleterious effects on numerous organ systems[3,4]. Some immune mediators, such as tumor necrosis factor-α (TNF-α), appear to be necessary

in responding to injury and promoting survival in blunt trauma patients and experimental animals[5]. However, elevated IL-6[6] and IL-10[7–9] in trauma patients are statistically associated with higher morbidity and mortality, and recent studies employing machine learning have implicated the type 3/Th17 response in these adverse outcomes[10–12]. These immune-inflammatory effects, in turn, likely cause or exacerbate multiple organ dysfunction syndrome (MODS) and other complications such as nosocomial infections following hemorrhagic shock[13]. However, the link between inflammation and clinical outcomes is not linear, and it is difficult, despite statistical

---

association, to link the actions of individual inflammatory mediators to clinical outcomes in the setting of T/HS[3].

An integral part of treatment for acute hemorrhage has recently been termed "damage control resuscitation" (DCR)[13]. This approach seeks to identify hemorrhagic shock at the earliest possible time, replace shed blood with hemostatic blood products and intravenous medications, and stop the spiral of coagulopathy and ongoing hemorrhage as quickly as possible[14]. A recent study demonstrated that prehospital plasma resuscitation improves survival following trauma[15]. It is hypothesized that early plasma resuscitation may modulate inflammatory and endothelial cell responses to injury[16], and recent work has associated survival with the necessity to normalize key coagulation protein concentrations[17]. Yet the complexity of T/HS-induced immune-inflammation and MODS, combined with the emergent and sporadic nature in which trauma often presents, makes clinical trial design both challenging and costly thus presenting a major translational barrier[18].

Mechanistic computational models based on ordinary differential equations (ODE) calibrated with data on trauma patients have yielded individual- and population-level insights into human T/HS and have enabled the assessment of novel therapies in silico[19]. Given that the development and regulatory approval of such therapeutics is not done in silico but typically involves studies in large animals that imperfectly mimic human disease, a core remaining challenge centers on linking granular pre-clinical data to data from patients[20], especially given the massive dimensionality of the response space of critical illness predicted by mechanistic models[21–23].

In the present study, we seek to utilize mechanistic computational modeling to bridge physiologic, inflammatory, and clinical aspects of T/HS in large-animal pre-clinical studies (in which early responses to experimental perturbations are assessed in very granular detail under controlled conditions but outcomes are limited to several hours after the insult) with human clinical studies (in which early data are sparse and inter-individual variability is high). Our model accurately predicts physiologic, inflammatory, and laboratory measures in large animals and humans with T/HS. In humans with T/HS our model correctly categorizes outcomes (survival vs death) and time of death. Further in silico simulation suggests resuscitation with plasma and red blood cells outperforms resuscitation with crystalloid or plasma alone, and that earlier plasma resuscitation reduces injury severity and increases survival time. Mechanistic mathematical modeling may serve as a translational bridge from pre-clinical to clinical studies in trauma-associated hemorrhagic shock and we suggest this represents a generalized approach to complex diseases.

## Methods
### Ethics and data analysis
Study oversight and ethical approval was provided by the Human Research Protection Office of Research Protections of the U.S. Army Medical Research and Materiel Command. A Cooperative Research and Development Agreement enabled sharing of animal and de-identified human data among participants for analysis. Statistical analysis was performed using R 4.0.3 (R Foundation for Statistical Computing, Vienna, Austria. https://www.R-project.org/).

### Mathematical model development
A three-compartment ordinary differential equation model was developed via expansion of a previous model of the inflammatory response to trauma[19]. This novel model consists of lung, tissue, and blood compartments, with an epithelial cell (EC) barrier between blood-lung and blood-tissue (Fig. 1). Migration occurs only from blood to lung or tissue; there is no reverse migration back into the blood and no migration from lung to tissue. Our model also accounts for inflammation and coagulation (Supplementary Figs. 1-2). Inflammatory mediators include monocytes (Mo), neutrophils (Nu), and pro- and anti-inflammatory cytokines (TNF-$\alpha$, IL-1, IL-6, IL-10, nitric oxide [NO]). An initial trauma, quantified by the injury severity score (ISS), initiates the model, perturbing it from the healthy steady state. Trauma activates Mo and Nu in all three compartments, and ECs on the two compartment boundaries. Activated Mo and Nu can migrate from the blood

to the lungs and tissue. Activated cells produce pro or anti-inflammatory cytokines and NO/iNOS. Pro-inflammatory cytokines promote the activation of additional cells. Anti-inflammatory cytokines inhibit these activation processes. IL-10 inhibits iNOS (which produces NO) production via all cell types. Nitric oxide reduces blood pressure (BP); in the model, NO levels above baseline result in lower BP, while NO levels below baseline increase BP. The main clinical output of the model is damage leading to death. We define damage as a function of BP, $O_2$Sat, IL-6, and the injury burden:

$$bp_{damage} = k_{damage_{bp}} * fm\left(\max\left(bp_{damage_{thresh}} - bp, 0\right), x_{damage_{bp}}, 2\right) \tag{1}$$

$$O_2Sat_{damage} = \frac{k_{damage_{O_2Sat}} * \max(O_2Sat_{damage_{thresh}} - O_2Sat, 0)}{\max\left(O_2Sat\right)} \tag{2}$$

$$il6_{damage} = k_{damage_{il6}} * fm\left(\max\left(il6 - il6_{damage_{thresh}}, 0\right), x_{damage_{il6}}, 2\right) \tag{3}$$

$$trauma_{damage} = \frac{k_{damage_{trauma}} * trauma}{\max(ISS)} \tag{4}$$

$$Damage = bp_{damage} + O_2Sat_{damage} + il6_{damage} + trauma_{damage} \tag{5}$$

where $fm(v, x, \text{Hill}) = \frac{v^{\text{Hill}}}{v^{\text{Hill}} + x^{\text{Hill}}}$, $\max(O_2\text{Sat}) = 98$, and $\max(\text{ISS}) = 75$. Damage-associated molecular pattern molecules (DAMPs, a byproduct of the initial trauma) promote the activation of the tissue-blood ECs. Activated lung-blood ECs inhibit oxygen transfer from the lung to the blood, resulting in low $O_2$Sat which feeds into damage.

The model was expanded to account for hemorrhage and resuscitation (Supplementary Fig. 2a-b). The coagulation module includes platelets, red blood cells (RBCs), inactive and active pro- and anti-coagulation variables, and clot formation. Trauma initiates the coagulation cascade, which is self-regulating, and leads to the formation of active coagulation factors (active procoag), which combine with platelets to form clots. Additionally, trauma enhances the degradation rate of the clots, as a mechanism to represent fibrinolysis. Active pro-coagulation factor (active procoag) promotes the activation of Mo, Nu, and ECs, linking coagulation to inflammation. IL-6 activates coagulation (inactive procoag to active procoag), further representing the known cross-talk between inflammation and coagulation. This model accounts for dynamic blood volume and blood pressure: trauma causes bleeding, which reduces the blood volume and lowers BP. Low blood pressure reduces the rate of bleeding, while elevated blood pressure, which can occur due to therapeutic infusions, increases bleeding. Low blood pressure also contributes to damage.

This expanded model serves as a platform for modeling therapeutic inputs, including resuscitation fluids and blood products. The model supports crystalloids and colloids, packed RBCs, platelets, and plasma (fresh frozen (FFP) and freeze-dried (FDP)). All infusions increase the blood volume (and thereby blood pressure). Ventilation is crudely represented: ventilation improves lung function, thereby decreasing $O_2$Sat damage. If a patient receives ventilation, this begins at hospital admittance. Time of death serves as a major clinical output of the model; passing a threshold in the value of the AUC of damage (276.53) triggers death. A narrative explanation of the model, model equations, definitions, parameters, and initial conditions are provided in Supplementary Notes,1-5 and Supplementary Data 1, and the model code has been made publicly available[24]. A total of 33 parameters were fit by minimizing a weighted least squares objective using a sequential Monte Carlo method. Factors influencing uncertainty during model fitting included degrees of freedom (range zero to ten for measured analytes), non-linearity of the model, and parameter estimation on an individual basis vs cohort basis.

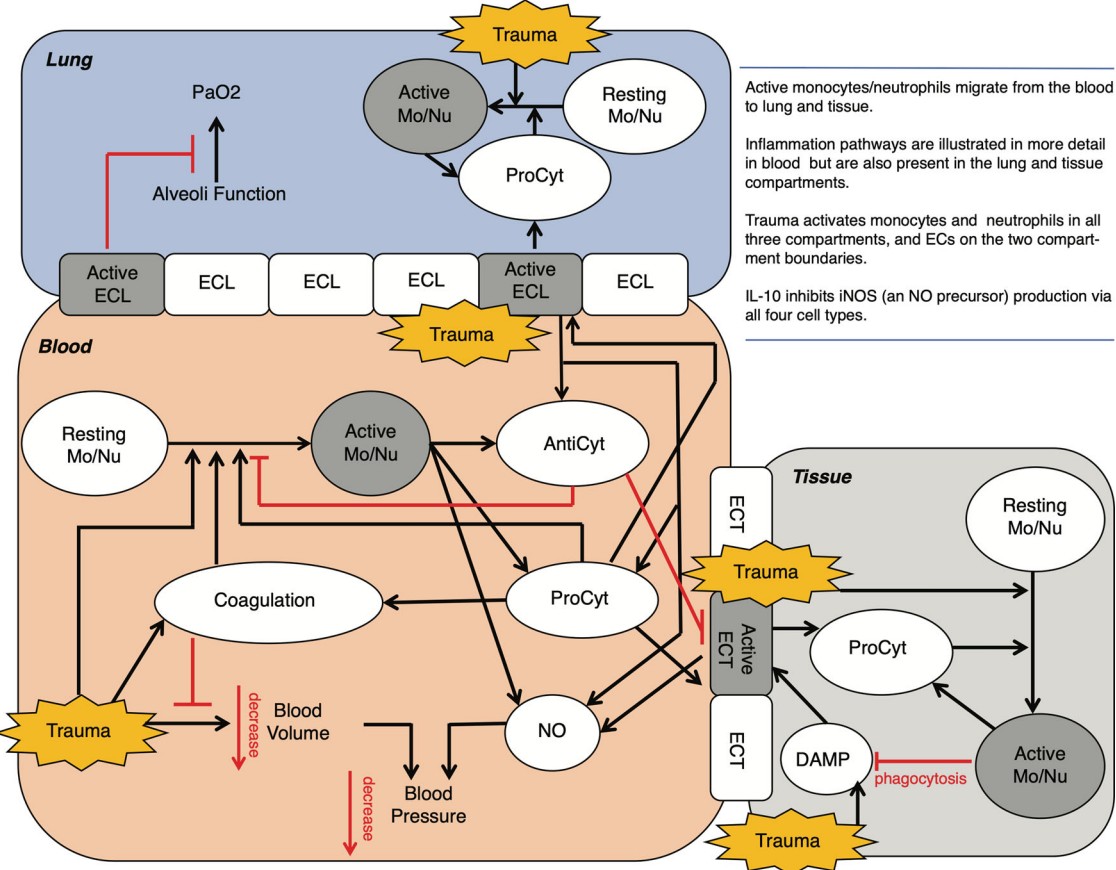

**Fig. 1 | Three-compartment ODE model.** The ordinary differential equation model used in Brown et al. [19] was modified to include aspects of the coagulation pathway. This model captures both the physiologic and biochemical response to trauma and predicts outcomes at the individual patient level. Black arrows signify an increase in effect, while red bars indicate an inhibitory effect. AntiCyt, anti-cytokine (anti-inflammatory) effect; DAMP, damage-associated molecular patterns; ECL, epithelial cell in the lung compartment; ECT, epithelial cell in the tissue compartment; Mo/Nu, monocyte/neutrophil; NO, nitric oxide; ProCyt, pro-cytokine (pro-inflammatory) effect.

## Porcine T/HS modeling

Data were obtained from Spoerke et al. in which all experimental procedures were done in accordance with the guidelines of the Institutional Animal Care and Use Committee at Oregon Health and Science University and the US Army Institute of Surgical Research[25]. Briefly, female juvenile Yorkshire crossbred swine were subjected to femur fracture with a captive bolt gun, cooled to 33 °C, then underwent controlled hemorrhage (60% of estimated blood volume), followed by 30 min of shock. Animals were then infused with isotonic sodium chloride solution, 0.9%, at volumes three times the controlled hemorrhage volume, to induce acidosis and coagulopathy. Subsequently, animals were subject to a Grade V liver injury, followed by 30 seconds of uncontrolled hemorrhage. Animals were then randomized to one of four different treatment arms: FFP only, FDP only, FFP + RBC in 1:1 ratio, or FDP + RBC in 1:1 ratio (four animals per arm), at rate of 50 mL/min, with an infusion volume equal to the blood removed during the controlled bleed. Four hours of monitoring took place before the animals were sacrificed. Laboratory measures were performed using standard techniques, and cytokine levels were quantified with enzyme-linked immunosorbent assays. Of the 32 animals in the original study, 16 were monitored with sufficient granularity to qualify for inclusion in this modeling study.

## Mathematical model assumptions and set-up

The mathematical model assumptions and set-up for the porcine T/HS protocol are illustrated in Fig. 1. The following assumptions were made for the ISS for the pig procedures: Line placement (ISS = 1), femur fracture/controlled blood (ISS = 15), and liver injury (ISS = 9). The $t_{initial}$ parameter is assumed to be 10 min prior to the femur fracture triggered by the placement of the monitoring lines. The rationale for this time period is that at baseline, the model sits at a healthy, steady state, where pro-inflammatory cytokines must have an initial condition of zero. However, the baseline pro-inflammatory cytokines do not have initial values of zero in the animal data set. This 10-minute buffer period, with a very small ISS of one representing line placement, allows the model to maintain a healthy initial steady state, but still reach the first reported cytokine data points by rising during the first 10 min.

Spoerke et al. [25] found little difference between FFP and FDP; so, for our purposes, we considered FFP and FDP both as "plasma" thereby collapsing the unique treatment arms from four down to two (Fig. 2). The 1:1 dosage in the packed RBC arms is replicated in the model by giving 1/4 of the total dose as packed RBC, then 1/4 as plasma, then 1/4 as packed RBC again, then the last one as plasma again. In the plasma-only arms, a single plasma dose is applied. However, note that in the mathematical model, the FFP and FDP infusions provide an identical amount of coagulation proteins. This is supported by published data[26]. For this model, we assume that the coagulation factors are reduced by 15% in both FFP and FDP due to the freeze/thaw or lyophilization process, respectively[25].

## Coagulation factor data

The mathematical model contains generalized species for pro-coagulation and anti-coagulation (inactive/active for each). Therefore, the data for pro-(FII, V, VII, VIII, IX, X, XI, XII) and anti- (ATIII, PC) coagulation species were compiled into groups, by taking the average value across pro- (FII, V, VII, VIII, IX, X, XI, XII) and anti- (ATIII, PC). We assume that the available

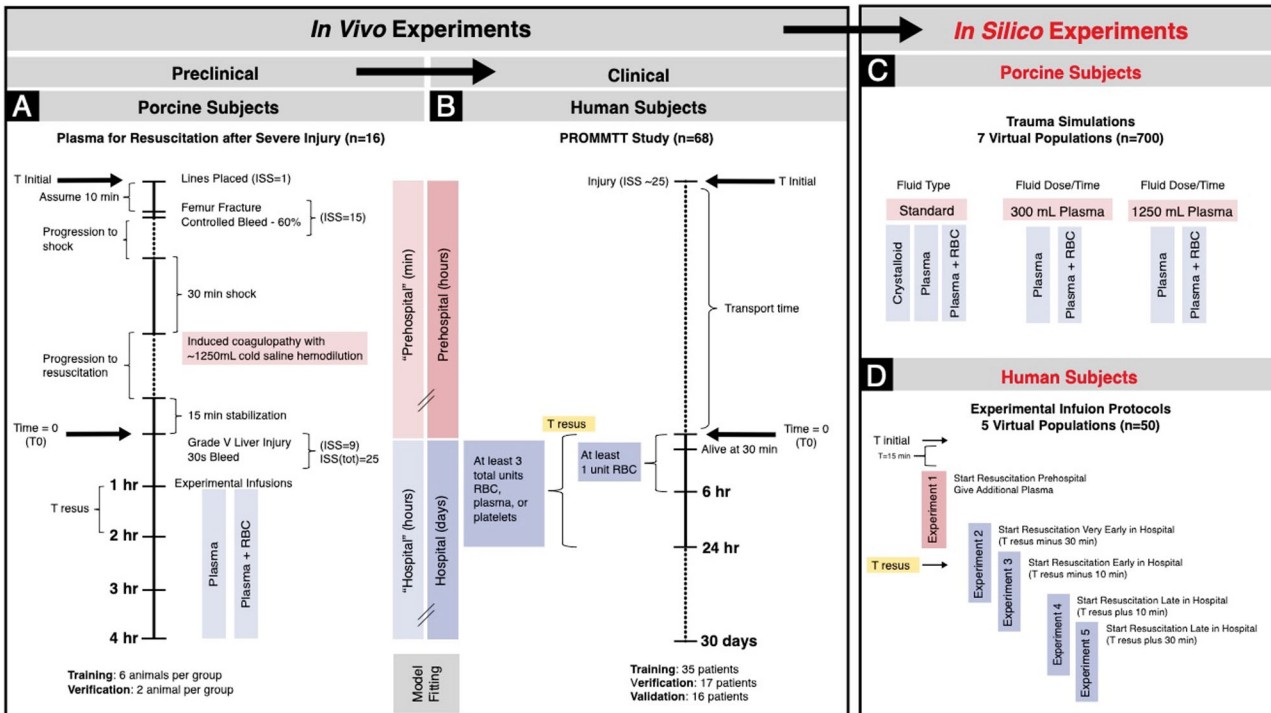

**Fig. 2 | Overview of model calibration and verification. A** Porcine data were obtained from Spoerke et al.[25]. **B** Human data were obtained from the Prospective, Observational, Multicenter, Major Trauma Transfusion (PROMMTT) Study[27]. **C** Trauma simulations for 7 virtual animal populations (100 animals in each group). **D** Experimental infusion protocols for 5 virtual human populations (10 in each group). RBC, red blood cells.

data represents the active and inactive forms of these factors summed together.

### Simulating bleeding
In the controlled bleed step of the actual animal experiments, the bleed rate was theoretically fixed, x/min (though x was unique per pig). However, in the mathematical model, the bleed rate is dynamic. Bleeding rates were tuned to blood loss volume targets from the animal data: heuristics were used in fitting to request that roughly the same amount that is bled in the experiment is also bled in the simulation, with similar timing to the real experiment. Thus, the reported bled volume is achieved in the model, however it is via a dynamic rate. The rationale is that in humans, in a trauma setting, the rate of bleeding will not be fixed or controlled, and the volume of blood lost is generally unknown. Having a fixed bleeding rate in the model would rob the model of its chance to tune the bleeding parameters, which is critical for moving on to the human patients, where the bleed rate is not constant or controlled.

### Fitting and verification with porcine data
Measured parameters for the animal model included TNF-α, IL-6, Mo, Nu, platelet, RBC, BP, and generalized variables for pro- and anti-coagulation. We fit to six out of eight pigs per arm ($n = 12$ total with $n = 6$ plasma-only and $n = 6$ plasma+RBC); two pigs per arm was reserved for verification ($n = 4$ total with $n = 2$ plasma only and $n = 2$ plasma+RBC). Data from all pigs were fit simultaneously, with the fit parameters divided into a large set of global parameters for all test subjects, and a small set of local parameters specific to individual subjects to elicit the differences observed among individuals. The types of parameters involved here include the following: 1) Fixed parameters directly from the pig data (such as initial blood volume), 2) fixed parameters from literature (such as cell death rates), 3) parameters that should be globally fit (includes most parameters), and 4) a small set is fit per pig, for per pig variability. Global parameters for death and damage were not fit in this stage (there is no death during the pig experiment, and death and damage do not feed back into the model). Model performance was verified

using two animals in each treatment arm that were not included in the model fitting stage. The intervention schedule and baseline characteristics of the individual pigs were used, and local parameters were fit, while the analyte trajectories were predicted using the locked global parameters from the fits.

### Simulating virtual porcine populations
In silico experiments were performed to assess the outcomes with alternative resuscitation strategies (Fig. 2). We sampled uniformly from the range of values found for locally fit parameters to generate virtual populations for different theoretical arms (protocols). The experimental protocol was not followed as closely on a persubject basis as in the individually tailored pig scenarios from the data, in that infusion volumes are not calculated based on blood volume here. Instead, all values were chosen from the range seen in the fitting. All experimental populations were generated to have 100 unique members to sample a meaningful portion of the state space with these simulations balanced against using a number of animals in which our findings could reasonably be verified. Seven virtual populations were grouped by pre-liver-injury protocol: 1. Administer the pre-liver-injury fluids protocol from the original experiment, then after liver-injury, administer: a) Fluids instead of plasma (volume equal to that of the original plasma infusion); b) Plasma-only (from the original experiment); or c) RBC + Plasma (from the original experiment). 2. Replace initial fluids with one unit (approximately 300 mL) of plasma, then after liver-injury, administer: a) Plasma-only (from the original experiment); b) RBC + Plasma (from original experiment). 3. Replace initial (nonleaked) fluids with an equal volume of plasma (approximately 1250 mL), then after liver injury, administer: a) Plasma-only (from the original experiment) or b) RBC + Plasma (from the original experiment).

### Human trauma modeling
The Prospective, Observational, Multicenter Major Trauma Transfusion (PROMMTT) study was an observational study designed to analyze the relationship between the timing of transfusions during active resuscitation and patient outcomes in ten Level 1 trauma centers in the United States[27]. It

was registered as ClinicalTrials.gov Identifier: NCT01545232, and Institutional Review Board was obtained at each study site with second-level review and approval provided by the US Army Human Research Protections Office. All relevant ethical regulations were followed. Waiver of informed consent was requested based on 1) minimal risk for this observational study and 2) high risk of refusal based on prior work. All participating sites approved this request except for the University of Washington, which required delayed consent from survivors. Additional data analysis was requested as part of the initial IRB application and was approved at all sites. Adult trauma patients surviving for 30 min postadmission, were transfused with at least one unit of RBC within six hours of admission and at least three total units of RBC, plasma, or platelets within 24 hours ($n = 905$). Detailed data were collected for RBC, plasma, and platelet infusions (timing and amount). Of these patients, we identified a cohort with complete, physiologically plausible data for all parameters of interest for use in our mathematical model ($n = 68$).

### Fitting, verification, and validation with PROMMTT data

Of the 68 PROMMTT patients suitable for modeling, 35 individual patients were selected for model training: nine of whom die in the first six hours, 26 survive at least 6 hours. The model was fit to the first 6 hours of patient data. $t_{initial}$, which is less than zero, is the time of injury, t = 0 is hospital admission, and the simulation proceeds to a max of t = 360 min if the patient survives that long. The fits are 1:1, with each virtual patient designed to fit to one real patient, with the actual reported infusion protocol for the real patient applied to the respective virtual patient. Fit data include $O_2Sat$ (a single data point at the start of hospital care), blood pressure (mean arterial pressure; zero to four data points are available per patient), platelets (a single data point at the start of hospital care), and time of death. Heuristics (assessed until death is triggered or the 6-hour cut-off) were also applied: Cytokine ($\alpha$) peaks <10,000 pg/mL; BP min/max of 20/250 mmHg; NO max of 1000 μmol/L, minimum bleed of 30% of healthy state blood volume (calculated per patient). Scenario definitions were created for the time of injury, ISS, initial blood volume, and interventions (mechanical ventilation, vasopressors, crystalloids, RBC, platelets, and plasma). Known baseline characteristics and intervention schedules were locked for each individual patient. Global parameters (other than damage/death) were locked, having been set via the pig fits. There is one exception to this rule: there was no NO data in the pig study fits, and all pigs were subject to the same trauma protocol. Given the possible differences in initial conditions between humans and pigs, it was unclear how this system would react to the variability introduced by the human data. Therefore, we re-tuned the following parameters as new global parameters, to be locked during validation/predictions: $K_{no-ma}$, rate of active monocytes making NO; $K_{no-ep}$, rate of active EPs making NO; $K_{no-inos}$, rate of iNOS making NO; $K_{bloodpressure-no}$, strength of NO on affecting blood pressure. Individual patient parameters were open for fitting. Damage and death parameters were open to fitting; these were fit globally and are locked from this fit. The model was fit to $O_2Sat$, BP, platelets, and time of death (or lack of death within the time frame of interest).

As in the pig-data fitting, model verification used a half-fit-half-predict approach: the global parameters from the fit stage were locked in verification, but the local parameters were still allowed to vary to fit the data. Again, intervention schedules were known, and local parameters were fit, but global parameters (this time including death and damage) were locked. 17 patients (eight die, nine survive) - separate from the fitting group - were initially used in verification. There were 16 additional patients (seven die, nine survive) who were subsequently used for final model validation in a "true hold-out" approach - no changes could be made to the model at this point.

### Death parameter

High IL-6, low $O_2Sat$, low BP, and trauma all contribute to the damage parameter. Cumulative damage is tracked by the AUC_DAMAGE parameter. A global death threshold is set in fitting, and when AUC_DAMAGE exceeds this threshold, death is triggered. However, even if death is triggered,

the model simulation continues out to 6 hours although the trajectory after death is triggered remains moot. Read the final value of the model trajectory for the time-of-death analyte to know the final time to consider. When "predicting" death, it is important to note that we are still fitting to last time alive (LTA), this is not a pure prediction.

### Simulated experimental infusion protocols

Each simulated experiment was run using ten of the patients from the fitting set (five survivors and five non-survivors, all of whom received at least one plasma infusion), and compared per patient to their original trajectory, which used their true in-hospital infusion schedule (Fig. 2). There is no clinically meaningful distinction between FDP and FFP in the model as supported by literature (i.e., the principal benefit of FDP is logistic in that it is immediately available without cold storage); so all plasma infusions were considered equivalent for these experiments[26].

**Experiment 1**. Administer an additional plasma infusion at an earlier time. The infusion was given 15 min postinjury (while still in the ambulance/in-transit) for all patients in this set. The volume and duration of this infusion was calculated using selected patients' average real first plasma infusion.

**Experiment 2**. Administer plasma infusion very early in the hospital-based resuscitation by shifting all infusion times to 30 min before baseline resuscitation start time ($T_{resus}-30$).

**Experiment 3**. Administer plasma infusion early in the hospital-based resuscitation by shifting all infusion times to 10 min before baseline resuscitation start time ($T_{resus}-10$).

**Experiment 4**. Delay the plasma infusion by 10 min after baseline resuscitation start time ($T_{resus}+10$).

**Experiment 5**. Delay the plasma infusion by 30 min after baseline resuscitation start time ($T_{resus}+30$).

### Reporting summary

Further information on research design is available in the Nature Portfolio Reporting Summary linked to this article.

## Results

### Study overview and process flow

We generated a three-compartment ODE model, modified from prior work[19], consisting of lung, blood, and tissue components with variables for inflammatory cells and mediators, coagulation factors, fibrinolysis, trauma, hemorrhage, and dynamic blood volume and blood pressure. The ODE model also contains therapeutic inputs for resuscitation fluids and blood products (Fig. 1). Supplementary Figs. 1-2 depict the key influences on the "damage" parameter and provide greater detail on the key modules of, and interactions in, the ODE model. Model outputs informed tests of simulated therapy strategies on virtual animal populations and a subset of the human patient data from which the model was developed. To generate virtual animal populations, we sampled parameters uniformly from values found in fitting and then tested variations in the trauma resuscitation in silico (Fig. 2). For the human simulated experimental protocols, data from $n = 10$ patients (five survivors and five non-survivors all of whom received at least one unit of plasma) were fit on a 1:1 basis (Fig. 2).

### Model calibration and verification: linking porcine and human data

Model parameters were tuned to data from pigs subjected to femur fracture and controlled hemorrhage followed by hemodilution and a severe liver injury (Fig. 2A). The model was calibrated on data from animals ($n = 12$) resuscitated with plasma alone or plasma and red blood cells (RBC). In this portion of the study, model parameters were tuned to

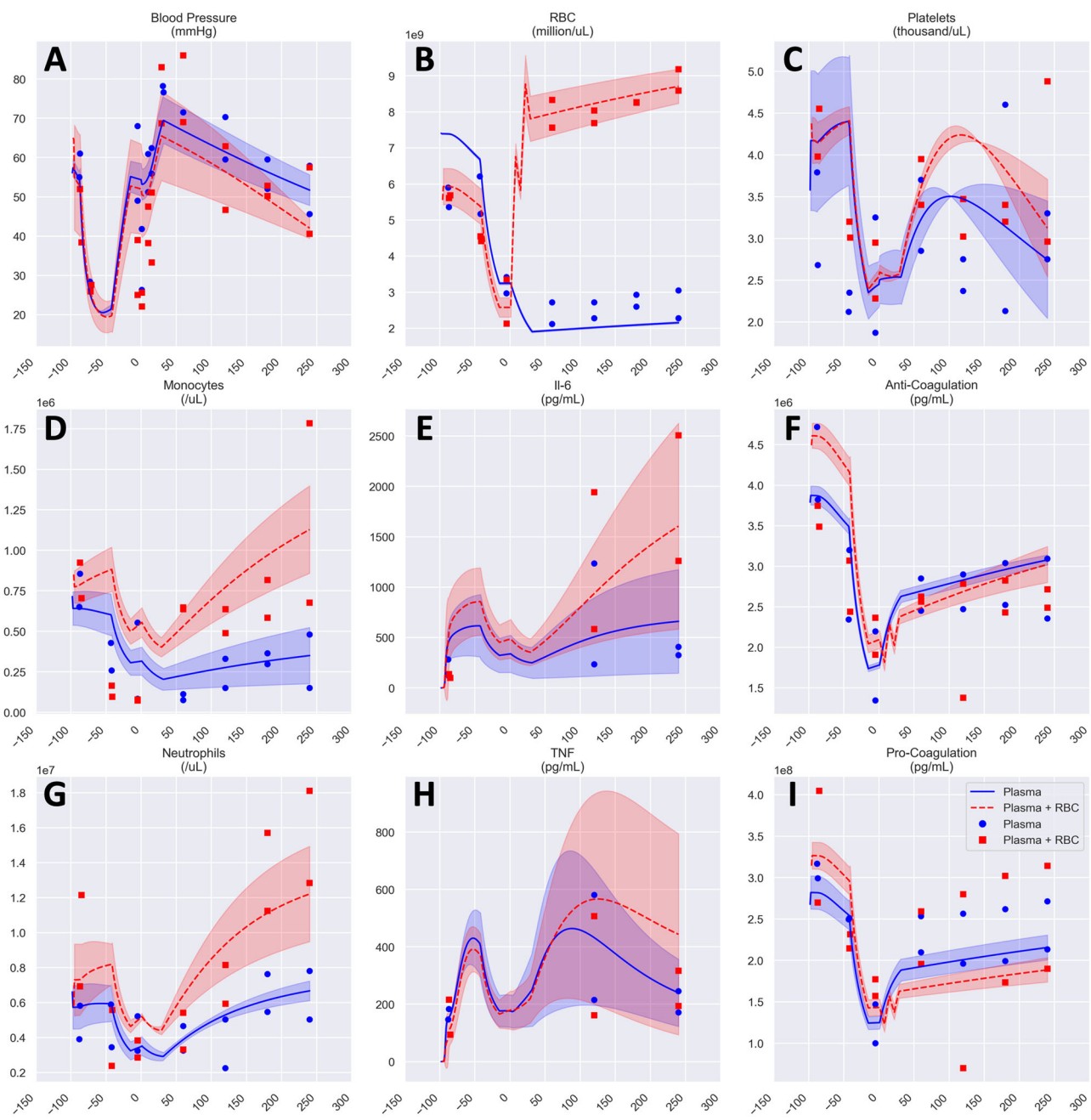

**Fig. 3 | Model fit verification for porcine subjects. A–I** Comparison of different resuscitation strategies in an animal model ($n = 2$ in each group). Experimental data shown as symbols, lines represent model fitted means, and shaded areas represent the standard error of the fitted mean.

data for blood pressure, circulating TNF-α, IL–6, monocytes, neutrophils, platelets, RBC, and coagulation factors. Additionally, computational heuristics were used to match the known blood loss volumes and to stay within a realistic physiologic range for IL-1β and IL-10. Primary simulated outcomes included blood loss and a composite "damage" metric in survivors that was composed of components including trauma, high IL-6, low oxygen ($O_2$) saturation ($O_2$Sat), and low BP (see Supplementary Figs. 1-2). Supplementary Fig. 3 depicts the results of the model calibration process in which data on hemodynamic, physiological, and inflammatory parameters from pigs subjected to T/HS and treated with FFP or FDP with or without packed RBC were used to tune model parameters. In agreement with the literature[26], resuscitation with plasma alone or plasma with packed RBC was essentially equivalent with respect to physiological and inflammatory parameters.

Model performance was verified by comparing model outputs to known results in holdout data from additional animals ($n = 4$). For these verification studies (Fig. 2A), global parameters from the model fitting stage were locked and known intervention schedules were applied. To maintain an initial healthy steady state with non-zero initial pro-inflammatory cytokine levels, local initial conditions were allowed to vary over the span of ranges observed in the fitting data set. Other locally fit parameters varied over the same ranges as in the animals used for fitting. The model predicted these fit targets accurately in the verification cohort based on visual inspection, regardless of the resuscitation strategy employed (Fig. 3).

A similar model calibration/verification approach was then applied to human data from the Prospective, Observational, Multicenter, Major Trauma Transfusion (PROMMTT) study[27] (Fig. 2b). For this part of the work, the model was trained using data from 35

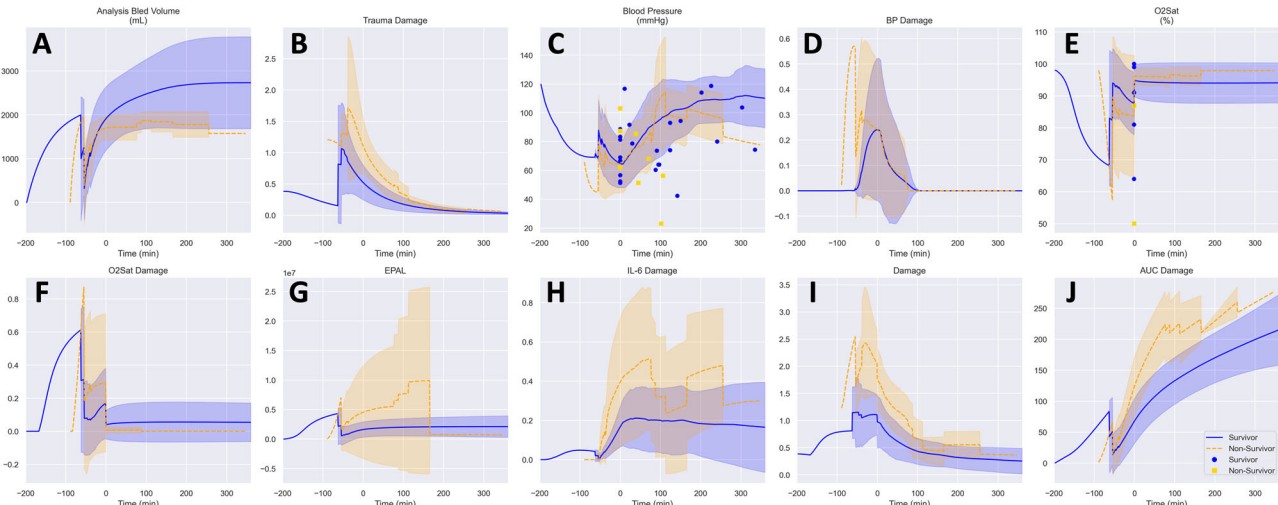

**Fig. 4 | Model validation in humans.** Comparison of survivors ($n = 10$) and non-survivors ($n = 6$) using models fit to human data from the PROMMTT Study[27] for systolic blood pressure (**C**) and oxygen saturation ($O_2$Sat, **E**). All other parameters including bled volume (**A**), EPAL (**G**) and all damage values (**B, D, F, H, I, J**) are derived values. Experimental data shown as symbols, lines represent model fitted means, and shaded areas represent the standard error of the fitted mean.

**Table 1 | Clinical measures and model performance**

| Measure | Survivor $n = 10$ | Non-Survivor $n = 6$ | $p^*$ |
|---|---|---|---|
| Model Error | 0.94 ± 0 | 1.01 ± 0.12 | 0.175 |
| SBP (mm Hg) | 96 ± 15 | 93 ± 29 | 0.827 |
| $O_2$Sat (%) | 93 ± 6 | 95 ± 5 | 0.646 |
| Bled Volume (mL) | 2,733 ± 1,047 | 1,718 ± 295 | 0.015 |
| EPAL | 1.82E + 06 ± 1.27E + 06 | 5.02E + 06 ± 9.23E + 06 | 0.436 |
| SBP Damage | 0.04 ± 0.05 | 0.07 ± 0.08 | 0.470 |
| $O_2$Sat Damage | 0.07 ± 0.11 | 0.04 ± 0.06 | 0.517 |
| IL-6 Damage | 0.18 ± 0.13 | 0.50 ± 0.36 | 0.077 |
| Trauma Damage | 0.23 ± 0.11 | 0.45 ± 0.15 | 0.011 |
| Total Damage | 0.51 ± 0.15 | 1.06 ± 0.36 | 0.011 |
| AUC Damage | 215 ± 57 | 442 ± 138 | 0.009 |

*AUC* area under the receiver-operator curve, *EPAL* count of active epithelial cells in the blood-lung barrier, *O₂Sat*, oxygen saturation, *SBP* systolic blood pressure, *SD* standard deviation
All values shown as mean ± SD
*two-sided t-test

patients (Supplementary Fig. 4), with verification in an additional 17 patients (Supplementary Fig. 5 followed by validation/prediction using data from 16 holdout patients (Fig. 4). The demographics, injury characteristics, and key outcomes of this PROMMTT sub-cohort are described in Supplementary Table 1 and survival probability is shown in Supplementary Fig. 6. The model calibration stage included an assessment of goodness-of-fit for data associated with survivors vs. non-survivors of T/HS. This analysis suggested that neither the measured physiological analytes and their predicted trajectories, nor the model-predicted trajectories for inflammatory mediators, clearly distinguished non-survivors from survivors during the calibration stage (Supplementary Figs. 4 and 5). Despite this, the calibrated model accurately categorized all survivors ($n = 10$) vs. non-survivors ($n = 6$) in the validation cohort based on differences in the damage (DAMAGE), trauma-specific damage (TRAUMA_DAMAGE), and cumulative damage (AUC_DAMAGE) variables (Fig. 4, Table 1); furthermore, for the non-survivors, the model predicted the time of death to within one minute (0.67 ± 0.82 minutes). The final model was re-parameterized to best reflect both porcine and human data and was fit within 9.8% of the original goodness of fit score.

**In silico experiments in porcine and humans suggest individually variable responses to distinct resuscitation strategies**

The final re-parameterized model was then used to conduct exploratory in silico studies in both animals (Fig. 2c) and humans (Fig. 2d) to assess the performance of plasma resuscitation timing and volume. Seven virtual animal populations ($n = 100$ animals each) were developed to examine the effect of fluid type, dose, and timing of plasma and RBC administration (Fig. 2c and Supplementary Fig. 7). These virtual simulations underscored the risks of under-resuscitation when pre-hospital time is prolonged (~45 min in this case) and predicted increased circulating inflammatory markers at four hours with large-volume pre-hospital resuscitation. Conversely, plasma and RBC resuscitation was predicted to result in increased clotting and reduced bleeding (Supplementary Fig. 8).

Subsequently, five in silico human experiments assessed the impact of giving plasma pre-hospital (Exp 1), starting the resuscitation in the hospital sooner by either 30 min or 10 min (Exp 2 and 3, respectively), or delaying resuscitation in the hospital by 10 or 30 min (Exp 4 and 5, respectively) (Figs. 2d and 5). Each in silico experiment was carried out using 10 of the patients from the calibration set (using all five non-

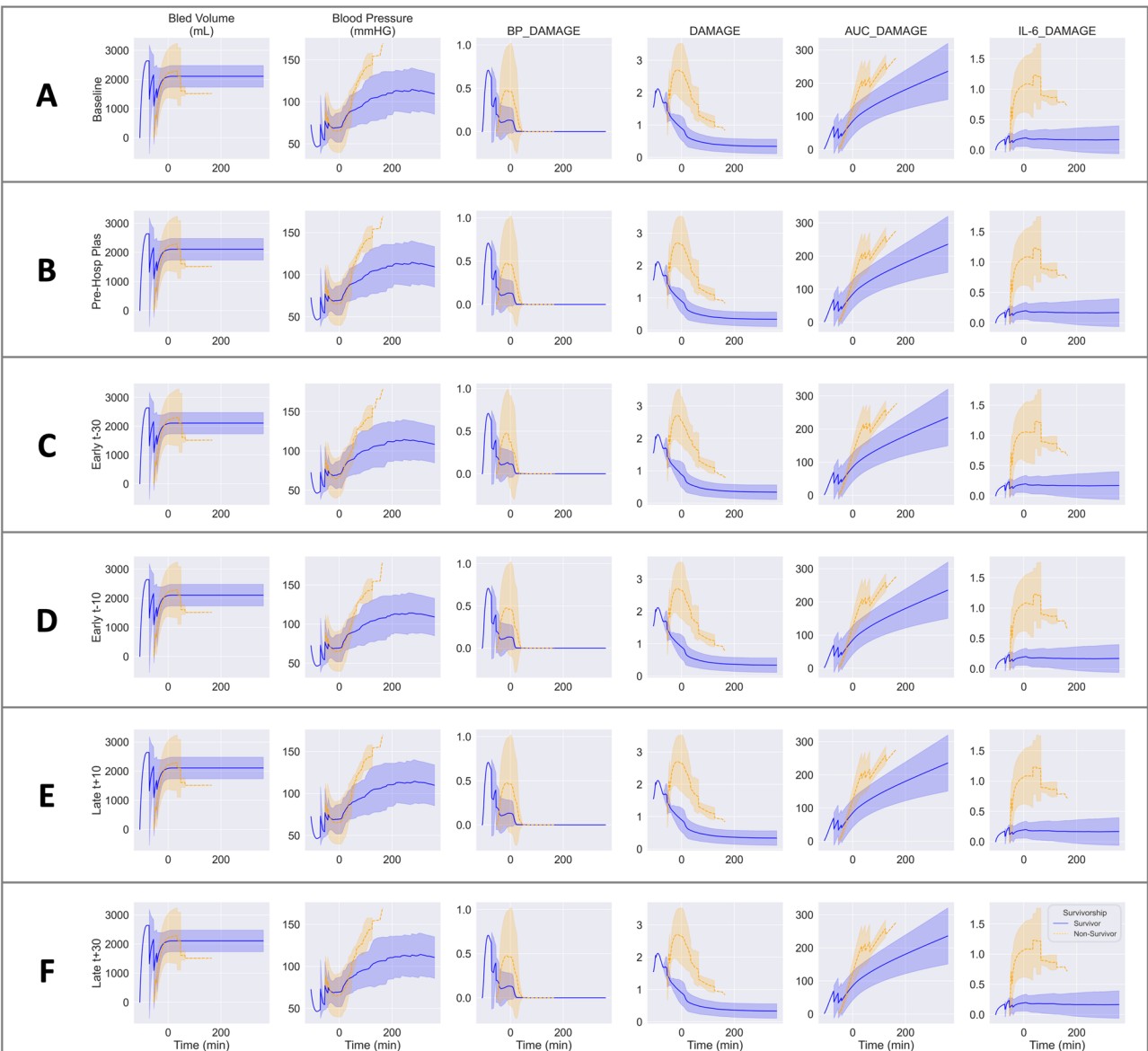

**Fig. 5 | In silico experimental infusions in humans. A** Baseline represents the model predictions for survivors ($n = 5$ patients) and non-survivors ($n = 5$) without modification to the resuscitation algorithm. In silico experiments represent predicted outcomes with variations in the resuscitation approach as detailed in Fig. 2D in the rows below baseline: **B** pre-hospital plasma given in addition to baseline resuscitation; **C** in-hospital resuscitation started 30 min earlier than baseline; **D** in-hospital resuscitation started 10 min earlier than baseline; **E** in-hospital resuscitation delayed by 10 min from baseline; **F** in-hospital resuscitation delayed by 30 min from baseline.

survivors who received at least one plasma infusion to reflect the high mortality in this population and five matched survivors). These simulations (Fig. 5b-f) were compared per patient to the original baseline (Fig. 5a), which used each patient's true in-hospital infusion schedule.

**In silico human experiment 1: prehospital plasma resuscitation**
In the PROMMTT study, plasma was administered, on average, more than an hour post-injury[27]. With this experiment, we assessed the effects of very early plasma resuscitation at a time designed to coincide with pre-hospital transport to test the potential benefits of the logistical planning required to deliver pre-hospital FFP or FDP[15,26,28]. Patients in this simulation (Fig. 5b) received an infusion of plasma 15 min post-injury, and the volume and duration of this infusion was calculated using the selected patients' average actual first plasma infusion. This experiment suggests that giving the first plasma infusion earlier (15 min

postinjury) increased bleeding by 5.2% ± 1.2% (mean ± standard deviation) over baseline bled volumes on a per-patient basis, but also increased survival time for non-survivors and decreased AUC_-DAMAGE for survivors (4.7% ± 4.5% and 2.7% ± 1.9%, respectively). In a clinical setting, the additional bleeding would be expected with the improved blood pressure that resulted in our modeling, and although the improved survival time and decreased damage were modest, these incremental benefits are consistent with recent clinical studies indicating early plasma has the greatest benefit for combat casualties[29] and polytrauma patients with TBI[15,30]. Our findings also suggest that a single intervention alone is unlikely to result in widespread, improved clinical outcomes for all patients at risk. Instead, we suggest this approach can allow us to refine the optimal target patients for each intervention while also assessing the effect of combination therapies, all warranting further exploration.

### In silico human experiments 2 and 3: early in-hospital plasma administration

We next simulated the administration of each plasma infusion earlier in the hospital course to assess the benefits of rapid recognition of hemorrhagic shock and immediate access to blood products in the Emergency Department[31]. We tested two cases by shifting infusion times by 30 or 10 min earlier (Figs. 5c and 5d, respectively). These experiments indicate that early pre-hospital resuscitation prolongs survival by several min without increasing blood loss among non-survivors. This prolonged survival would potentially allow non-survivors to proceed to operative intervention sooner, resulting in improved survival[32]. Furthermore, for survivors, early resuscitation decreases AUC_DAMAGE by a small amount. Based on analysis of these simulations, these benefits were mediated through lower levels of BP_DAMAGE and IL-6_DAMAGE by boosting blood volume and diluting IL-6 earlier.

### In silico human experiments 4 and 5: late in-hospital plasma administration

The last in silico experiments were designed to test the effect of delayed plasma intervention to simulate prolonged transport to the hospital or delayed recognition of hemorrhagic shock. These results showed minimal difference in outcomes: shifting the plasma infusions 10 to 30 min later (Figs. 5e and f, respectively) had a small detrimental effect on AUC_DAMAGE and survival time parameters but no effect on bleeding.

Overall, the five in silico experiments suggest that clinical outcomes following T/HS are sensitive to the timing of early infusions, consistent with recently published clinical trials[33]. Rapidly reversing hypoperfusion and maintaining healthy blood pressure appears to be the most critical aspect in mitigating overall damage and is even beneficial at the cost of extra bleeding. Our results support the benefit of early DCR with blood products and highlight the potential for personalized care in T/HS using this modeling-based strategy.

## Discussion

Trauma is the leading cause of death and disability for individuals under the age of 55 years in the United States[34]. Hemorrhage represents the leading cause of potentially preventable death in both civilian and military trauma patients in both the prehospital and in-hospital environments[35]. The subtleties of hemorrhagic shock and the effect of very slightly different levels of hemorrhage on the ultimate outcome of the individual have been appreciated for nearly a century[13]. Thus, identifying patients with occult hemorrhage at risk for hemorrhagic shock and initiating life-saving treatment like plasma and whole blood resuscitation early has become a major emphasis in trauma research. Early intervention in these cases is paramount to avoid the spiral of coagulopathy, acidosis, and hypothermia that carries a mortality of approximately 50%[36,37]. However, rapidly and accurately identifying patients in hemorrhagic shock who can be salvaged with timely intervention remains an unsolved clinical conundrum[13,14]. Scoring systems and statistical predictive models based upon early vital signs have been described[38–41]. Although these approaches are effective at identifying more obvious cases of profound shock, they generally miss those patients in shock whose compensatory mechanisms mask their otherwise tenuous physiologic state.

Mechanistic modeling, in contrast with pure data-driven and statistical models, may be able to tease out these subtleties at both the individual and population level[42–44]. Mechanistic computational models can represent the current state of biomedical knowledge at cellular and molecular scale appropriate for clinical translation[44,45]. Thus, we hypothesize that mechanistic modeling can accurately and reliably identify patients at risk for hemorrhage and death given limited initial data, thereby facilitating those patients at high risk of death from hemorrhage, even before shock becomes clinically apparent. However, there is a need to unify pre-clinical and clinical studies in order to realize the full promise of mechanistic computational modeling, which includes the potential for patient-specific simulations (i.e. "digital twins") and simulated populations (i.e. in silico clinical trials)[18,44,46].

While there have been successful prior demonstrations of data-driven modeling as a means of linking pre-clinical and human-level data[47], prior efforts utilizing mechanistic models of T/HS have focused solely on inflammation[19] or coagulopathy[48,49]. We have suggested previously that mechanistic computational modeling may hold certain advantages due to the explicit representation of biological mechanisms[50]. We therefore undertook the present study to demonstrate the value of stepwise development and refinement of mechanistic models as a means of linking pre-clinical and clinical data.

We based the present study on the mechanistic model of human trauma described previously[19]. This model was designed to recapitulate the inflammatory response to blunt injury and the effect on the whole organism. To include an assessment of the effect of severe hemorrhage, we added elements of the coagulation system to this model. We also allowed for inputs of different resuscitation fluids including crystalloid and various blood products. We retained the mortality prediction element of the former model using an integrated outcome termed "damage"[19], wherein small amounts of damage represent morbidity while unresolved damage indicates death.

We then examined this enhanced model in the context of an animal study with tightly controlled experimental conditions and reasonable initial conditions[25]. Modeling these animal experiments allowed us to tune the model to reflect various biomarker trends. After verifying appropriate performance of the model, we assessed outcomes in terms of damage and blood loss with different resuscitation strategies. This analysis (700 virtual subjects in seven different arms) demonstrated that the combined outcomes of blood loss and damage were minimized using a hemostatic resuscitation approach, whereas a crystalloid-only approach increased both blood loss and damage. This is consistent with recent prospective and secondary analyses, which demonstrate the additive benefits of early blood-component resuscitation[15,51,52].

Our model demonstrates that the plasma and RBC infusion results in increased clotting and reduced bleeding as compared with crystalloid fluid resuscitation. Crystalloid appears to have a dilutional effect. A previous in silico model built to study the dilutional effects of conventional component therapy versus whole blood in the management of massively bleeding adult trauma patients also demonstrated that prehospital blood product transfusion in place of crystalloid resulted in higher hemoglobin and fibrinogen concentrations and a lower international normalized ratio throughout the resuscitation regardless of the resuscitation strategy used[53]. While neither modeling approach assessed the effect of crystalloid on endothelial damage, it is possible that crystalloid resuscitation may also exacerbate the endotheliopathy of trauma[16]. Future mechanistic modeling work should also examine these possible effects.

These results informed our model for predicting outcomes in prospectively collected clinical data on patients at risk for death from bleeding[27]. In the PROMMTT study, which included 905 patients, there was a 25% mortality rate. Of the non-survivors, 42% had hemorrhage reported as a cause of death[27]. Our model demonstrated a high degree of accuracy in predicting both survival and time of death.

Finally, we varied the time and type of resuscitation in five separate in silico human experiments. In this integrated analysis, we demonstrated that moving resuscitation to the earliest time possible prolongs survival in nonsurvivors and decreases damage in known survivors. Thus, early resuscitation on a large scale may allow sufficient time to perform life-saving surgical hemostasis and may also lessen morbidity in survivors as has been seen in prehospital clinical studies[15].

Several limitations of the current model and our findings must be recognized. First, the coagulation system, the adaptive and maladaptive physiologic responses to hemorrhage, and the interplay of resuscitative efforts on all these systems represent a dynamic system of vast complexity. Thus, any attempt to model these incompletely understood and highly variable systems will necessarily represent an abstraction that, at best, approximates reality. As an example, the animal phase of model development assumed that subjects were relatively hyper-coagulable based on the literature[54]. However, the degree of hyper-coagulability and the necessary adjustments to the model for use in

humans were both unknown, potentially compounding inaccuracies due to partial quantification of the various elements of the coagulation system and the lumping of all coagulation factor levels into an aggregate measure. Furthermore, this iterative model assumes the fundamental importance of the lung, blood, and tissue elements of the model developed in our previous work. This model, however, does not account specifically for the neuroendocrine response to trauma and hemorrhage or for large endothelial surfaces like that of the small intestine that may contribute substantially to irreversible shock[13]. Furthermore, our approach to parameter fitting carries multiple potential limitations, including the use of some parameters from literature sources that may or may not apply, intrinsic non-identifiability of the real and model systems with associated uncertainty about the parameter estimates obtained by this process, uncertainty regarding possible outputs of the system, and the existence of additional, equally plausible model structures and/or parameter regimes that would fit the data equally well. The initial animal data used for model development did not include any early deaths from hemorrhage; thus, the ability of the model to accurately predict early death was not able to be specifically assessed at this stage of model development. The human data are also sparse (e.g., a single data point for platelets and for $O_2$Sat, and between zero and four-time points for BP). This does not account for the dynamics of these variables. There are many factors that affect BP in a clinical setting that are not accounted for in our model (e.g., stress/pain, surgery, medications, and others). Patients reported to have zero infusions were excluded, as it was assumed that the data was in fact missing. However, it is possible that there is partially missing infusion or maintenance fluid data, which will affect our accuracy. The current model is based in part on the assumption that the Injury Severity Score (ISS)[55,56] is a reasonably accurate marker of the level of injury. Because some outlier results were observed in the dataset (e.g., a patient with ISS of two who died, or an ISS of 75 who lived), we excluded patients with ISS < 5 or >70 to remove outliers. Finally, we did not assess the effect of human modeling results with and without the benefit of initial model development and training/verification on animals.

These limitations notwithstanding, our work demonstrates the value of an iterative, mechanistic approach to linking pre-clinical and clinical data via mechanistic computational modeling, leading to accurate predictions of uncertain physiologic outcomes and to testing different interventions in silico. Future applications of this approach should assess the generalizability of this approach. This will require collecting highly granular physiologic, laboratory, and interventional data on a relatively small number of subjects with a wide range of initial conditions (including injury mechanism, rates of hemorrhage, and time to initial treatment) and subsequent outcomes[57–60]. This modeling approach also has the capacity to evaluate outcomes from hemostatic medication interventions such as prothrombin complex concentrate[61,62] or vasopressin supplementation[63].

To further develop this capacity for performing in silico clinical trials, we suggest that the same sequence of steps used herein should be followed (Fig. 2). As noted previously[20,43,50], the performance of the computational model should be assessed with existing data from a small number of animals (ideally with a range of outcomes) after which any model adjustments are made before adapting it to human application. Then, data from a small number of human subjects can be fed into the model for validation, after which a full-scale interventional in silico trial can be conducted to determine population-level outcomes. Multiple iterations of the trial could be run to determine the ideal candidates for therapy as well as treatment specifics, such as the optimal dosing and timing for the therapy. Also, harmful or unnecessary therapies could be identified rapidly and eliminated from further consideration for clinical testing. Ultimately, we suggest that this workflow holds the potential to integrate pre-clinical and clinical data not only in the context of T/HS but is also applicable more generally to complex disease settings.

## Data availability
Large animal data for this study is from Spoerke et al.[25]. supplied to the corresponding author by Oregon Health & Science University, Portland, OR, USA. Human data is from the Prospective, Observational, Multicenter, Major Trauma Transfusion (PROMMTT) Study[27] supplied to the corresponding author by University of Texas Health Science Center at Houston, Houston, TX, USA. The data used for model training, verification, and validation and the modeling output reported in this study are not openly available due to reasons of sensitivity. Data are stored under controlled access at the University of Pennsylvania and are available from the corresponding author on reasonable request. All figure source data is publicly available in the Dryad Digital Repository at https://doi.org/10.5061/dryad.f4qrfj733.

## Code availability
The model code is available at https://doi.org/10.5281/zenodo.10595453.[24]

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

## Acknowledgements

The authors gratefully acknowledge the administrative support of Ms. Martha Brinson in preparing this submission. This work was supported by a grant from the US Army Medical Department (AMEDD) Advanced Medical Technology Initiative (AAMTI) executed by the Geneva Foundation to J.W.C., N.B., K.H., J.B.H., J.L.S., S.U., M.A.S., A.I.B., L.C.C., S.C.C., & Y.V.; by NIH grant P50-GM-53789 to Y.V.; and by NIH Postdoctoral Fellowship 5T32GM008516-25 to D.S.G.

## Author contributions

J.W.C. secured study funding, participated in experimental design, data interpretation, and study oversight, and contributed to writing the manuscript. D.G. participated in data interpretation and contributed to writing the manuscript. N.B. participated in data analysis and modeling, results interpretation, and contributed to writing the manuscript. K.H. participated in model development, experimental design, results interpretation, and provided research oversight. J.H. participated in model development, experimental design, results interpretation, and contributed to writing the manuscript. R.Z. participated in data analysis, and interpretation and manuscript writing. F.E. participated in data analysis and interpretation. Z.G. participated in data analysis and interpretation. R.N. participated in results interpretation and provided critical manuscript revisions. J.L.S. participated in results interpretation and provided critical manuscript revisions. J.B.H. provided clinical data, participated in results interpretation, and provided critical manuscript revisions. B.A.C. participated in experimental design, results interpretation, and provided critical manuscript revisions. J.J.N. participated in results interpretation and contributed to writing the manuscript. S.U. provided animal data and participated in results interpretation. M.A.S. provided animal data and participated in experimental design, results interpretation, and provided critical manuscript revisions. K.K.C., A.I.B., and L.C.C. participated in experimental design and provided critical manuscript revisions. A.J.B. provided insightful commentary on mechanistic modeling, clinical expertise on results interpretation, and critical manuscript revisions. E.E.F. provided clinical data, participated in experimental design, results interpretation, and provided critical manuscript revisions. S.C.C. and A.P.C. participated in experimental design, data interpretation, study oversight, and provided critical manuscript revisions. Y.V. participated in experimental design, data interpretation, and study oversight and contributed to writing the manuscript.

## Competing interests

Y.V. is a co-founder of and stakeholder in Immunetrics, Inc. All other authors have no competing interests to declare. The views expressed herein are those of the authors and do not reflect the official policy or position of the US Air Force, the US Army Medical Department, the US Army Office of the Surgeon General, the Department of the Army, the Department of Defense, or the US Government.

## Additional information

[1]Division of Traumatology, Surgical Critical Care & Emergency Surgery, Perelman School of Medicine at the University of Pennsylvania, Philadelphia, PA 19104, USA. [2]Department of Surgery, Uniformed Services University of the Health Sciences, Bethesda, MD 20814, USA. [3]Department of Surgery, University of Pittsburgh, Pittsburgh, PA 15213, USA. [4]Pittsburgh Trauma Research Center, Pittsburgh, PA 15213, USA. [5]Center for Inflammation and Regeneration Modeling, McGowan Institute for Regenerative Medicine, Pittsburgh, PA 15219, USA. [6]Immunetrics, now wholly owned by Simulations Plus, Pittsburgh, PA 15219, USA. [7]Department of Surgery, University of Alabama, Birmingham, AL 35233, USA. [8]Division of Acute Care Surgery, University of Texas Health Science Center at Houston, Houston, TX 77030, USA. [9]Department of Medicine, Uniformed Services University of the Health Sciences, Bethesda, MD 20814, USA. [10]Division of Trauma, Critical Care and Acute Care Surgery, Oregon Health & Science University, Portland, OR 97239, USA. [11]SeaStar Medical, Denver, CO 80216, USA. [12]Autonomous Reanimation and Evacuation (AREVA) Research and Innovation Center, San Antonio, TX 78235, USA. [13]US Army Institute of Surgical Research, Fort Sam Houston, TX 78234, USA. [14]Trauma and Acute Care Surgery, Department of Surgery, The University of Chicago, Chicago, IL 60637, USA. [15]Center for Systems Immunology, University of Pittsburgh, Pittsburgh, PA 15213, USA. ✉e-mail: jeremy.cannon@pennmedicine.upenn.edu

