## [Peer Review File · Communications Medicine]

Reviewers' comments:

Reviewer #1 (Remarks to the Author):

As someone who is very familiar with the good work previously produced by several of the authors, it was a pleasure to review this manuscript. The point of this paper is to develop a computational model that can predict and guide trauma patient outcomes and treatment from limited pre-hospital patient data when supplemented with more detailed animal-model data, collected in the immediate aftermath of severe injury. The work is soundly motivated, timely, and novel. This reviewer can appreciate the scope and complexity of the described model development, given the large number of associated parameters and the overall model size.

I have a number of suggestions to improve the overall clarity of presentation. At a high-level, this boils down to providing more technical detail at certain manuscript locations, as indicated below. Also, the validation process as described is a little unclear, at least in terms of visual comparisons with the actual data, and also in not fixing all of the parameters during the validation stage, which is what should be done.

Line-by-line comments:

Line 74: You may want to point out that evidence for an underlying rationale in support of this hypothesis was recently put forward, and it has to do with how a patient's protein concentrations normalize or not depending on what exactly is provided in the plasma. It seems that patients will survive if, and only if, key protein concentrations normalize. See Ghetmiri et al, npj Systems Biology, 2021.

Line 86: The two references are for sepsis (and by the same authors). A more apt, albeit older, mechanistic model of coagulation with massive dimensionality is Luan et al, Molecular BioSystems, 2010.

Line 101: I feel that some description about virtual populations are in order here, before going any further. How were the virtual populations generated? How representative are they, and what are the guarantees of the population statistics? For what reason was the virtual population generated: testing or updating the model? How do the generated population statistics (differing from the actual statistics) affect the goal of testing or model updates (i.e., what are the biases, if any?)

Line 123: "local initial conditions were allowed to vary over the span of ranges observed in the fitting data set" This is unclear in light of the previous sentence on locking (globally) vs the non-locking locally (next sentence), and also the emphatic distinctions you are drawing on verification (really, just continued model fitting) compared to validation. For this line 123, why were local initial conditions allowed to vary? At the end of the day, since this is holdout data, it makes more sense to check real vs. predicted outputs for fixed initial conditions, especially since $n=4$ (small number). Varying initial conditions will impact predictions and affect accuracy.

Line 125: "fit targets accurately" How accurately? What are the metrics? There are large ranges, so how good is this fit overall?

Line 126: Fig 3. Legends and figure panels should explicitly show and state experiment and

predicted.

Line 131: Fig 4. First, it is unclear what is being plotted, predictions or data. (I suspect just predictions, but this is not completely clear from the caption or the panel legends.) For validation, as is stated as the goal in the caption, both predictions and data should be shown for visual comparison. (I acknowledge that showing all data can be messy, but it is useful.) Because, if these are just predictions, then how do the lines show validation against actual? And if they are data (e.g., the shaded regions) vs. the model (e.g., the lines), then the caption doesn't make sense.

Line 138: "accurately categorized." How accurate? What are the metrics?

Line 142: This section is about linking porcine and human data. But after reading this section carefully, it wasn't clear to me what the linkage is. The first two of the three paragraphs in the section are about swine, the last paragraph is about human. Where is the paragraph about the linkage? For example, is the connection that some/all parameters in the swine-tuned model were scaled-up for, or turn out to be a scaled-up version of, the human-tuned model? Which parameters? Or is the connection that some output from the swine-tuned model fed into the human model? If so, which ones, why and how? Essentially, the linkage needs to be clearly articulated. This gets to the "integrate" in the title of the paper.

Lines 147-149: How were the virtual animal populations developed, how different are they, how do they examine the effect of therapies? You did seven different populations for each treatment, why not use one population and check each of the seven treatments on the same population?

Lines 154-156: Is it three or is it five human populations, based on Figure 2D and 5? Typo?

Lines 156-159: Simulation and original trajectories are not shown in the figure, so how to compare?

Line 160: It seems to me like the goal should be made really explicit. Are you planning to evaluate how much the porcine data negatively impacted the human prediction, since the human data is the same for training and predictions? It would be useful to see predictions with and without the porcine learning, to assess overall impact of the porcine data. This goes to the significance of the study.

Line 169: "over baseline." What is the baseline? Is it per patient, or is it the mean in the population?

Line 169: "5.2% plus/minus 1.2%" Somewhere near here, it needs to be confirmed that this is mean plus/minus 1 std dev?

Lines 169-171: The percent results are small. So how useful are these numbers? A remark needs to be added. Same for line 180, 184.

Lines 370-371: It is not clear what ODEs are used, or what type of solver was run to calculate the ODEs. The ODEs are not specified separately in the supplement as is typical for these sorts of papers (although it appears they are embedded in the code, but these equations are hard to read and parse). Initial conditions and parameters are made very clear. Having a description of the ODEs being solved in each compartment in the supplemental materials would be helpful.

Lines 409-412: There is no mention of tissue factor even though this is a key part of the extrinsic signaling cascade associated with trauma. Is tissue factor considered at all in the model, and if not, why was it chosen to be left out?

Line 410-412: Summing/averaging the coagulation factor data together is somewhat confusing. The factors are often measured in different units, and have a wide range of concentrations so often one factor is present at a much larger magnitude than another factor. Therefore, if these numbers are averaged and used as general pro-coagulant and anti-coagulant single measures then they might not account for smaller concentration changes occurring in individual factors, which may have a large impact on the overall cascade.

Lines 502-512: So if local parameters are being allowed to vary, while others are locked, then this is **not** validation (as in the subsection title). Earlier, you call this verification. In general, the whole manuscript needs to be really clear about what is validation: everything should be fixed, and you compare predictions against data that was not used for learning.

Lines 516,517: auc_damage is written differently than elsewhere in the manuscript.

Remark on author contributions: for a modeling paper, even if the model is large, this paper seems to have a rather large author listing.

Figure 5: same comments as were made for Figure 4. Also, Figure 5B the blood pressure graph scale changes, making it much harder to distinguish a difference between the patient's original trajectories (shown in Figure 5A) and the simulated change of getting plasma much earlier. The y-axis should be adjusted so the scale remains the same across all columns of the figure to make comparisons easier. Same comment for 5D.

Supp Figure 2: why do some graphs have marker points but other graphs only have lines? What do these points represent? Additionally, what are the units on the y axis of each graph?

Supp Figure 3 is missing the bottom half of the figure (parts F – J)

Remark on Title: The concept of a digital twin is still relatively novel and there is no universal definition just yet. However, it is generally accepted that a digital twin is more than a personalized model, since it allows for some sort of patient feedback (either in real-time, or delayed through measurements and periodic model inputs), to output updated patient trajectories. Since this model has no real-time patient feedback component, I would argue that it is more of a personalized model than an actual digital twin.

Reviewer #2 (Remarks to the Author):

I like the idea behind this effort, which was to "unify pre-clinical and clinical studies in order to realize the full promise of mechanistic computational modeling, which includes the potential for patient-specific simulations (i.e. "digital twins") and simulated populations (i.e. in silico clinical trials)". This is, however, not a novel concept/approach. Similarly, their conclusions that "resuscitation with plasma and red blood cells together outperformed resuscitation with crystalloid or plasma alone, and that earlier plasma resuscitation reduced both morbidity and mortality" are

not novel.

So, while I like the academic exercise, I don't see how this paper adds much to the existing literature. At best these integration efforts provide a rough approximation of the reality, that fails to capture the person-to-person variability. This is a major reason why interventions that work in well controlled animal studies fail (or are not as effective) in human trials.

Reviewer #3 (Remarks to the Author):

See attached. Contact handling editor if not accessible.

Brief summary of the manuscript

Hemorrhagic shock, the depletion of intravascular volume through blood loss to the point of being unable to match the tissues demand for oxygen, is investigated using a digital twin set-up based on animal/human data. Linking preclinical data in swine and clinical data from patients via a 3-compartment ODE model of inflammation and coagulation, the authors seek to identify intervention mechanisms. Their model predicts accurately measurements, such as time of death in patients, and is used to conclude that plasma and RBC infusion results in increased clotting and reduced bleeding.

Overall impression of the work

The use of mathematics and digital twins is an exciting area of research and it is fantastic to see an application into Hemorrhagic shock. Unfortunately, I find the details surrounding the mathematics insufficient to reproduce the work and to critically analyse the validity of the results presented by the authors. If the authors could provide more details about their model, and the methodologies they used, this would significantly support their manuscript and make this a very impactful piece of work. I would be happy to review a revised version of their manuscript.

Specific comments, with recommendations for addressing each comment

1. Mathematical modelling details: “Model parameters, initial conditions, and code files are provided in Supplementary Materials”, in my opinion the details provided in the supplementary materials are insufficient in their current form to accurately describe the model so that it can be (i) critically understood and (ii) reproduced. Could the authors please provide the list of ordinary differential equations, along with the meaning of the terms in the equation, motivation (from the literature or otherwise) for the base forms of mechanisms being modelled, details of mechanisms that were omitted and then how each of the parameters were estimated. In the Materials and Methods a lot of the assumptions are listed in terms of their biology but there is no reference to the modelling terms that represent them.
2. 3 compartment ODE: Further to my above comment, without reading Brown et al. it's not clear how Figure 1 depicts a 3 compartment ODE model. Usually, this phrase means there are only 3 ODEs, however, I think based on the simulations there are more? What do all the nodes and arrows represent in their figure in each of the ODEs? More details on the equations, exactly what each variable means etc is needed. Minor comment: caption should be able to summarise the figure without needing to read the text, more detail on what different coloured arrows mean, different coloured boxes etc is necessary.
3. Fitting algorithm: There is insufficient information on how the parameters were fit in the model and which parameters were fit. For example, what fitting algorithm was used? How was the uncertainty in the fitting evaluated? What confidence intervals were obtained? How many parameters were fit to how many data points. Also where are the images for the model fit to data?
4. Virtual individuals: How were the number of virtual individuals in each group chosen? E.g. why were there only 10 in each group for the humans? The authors say “we uniformly sampled from the range of values found for locally fit parameters to generate virtual populations for different theoretical arms” was this only for the swine virtual trial? Or also for the humans? but then why were only 10 humans in each group used? How did the authors confirm their virtual population was representative of a sample from their interval? Why did the authors not sample more human patients from an interval returned for the fitted data?

5. Motivation for swine and human in silico trials: I'm not sure I understand why both the pig and human data were used? Why not just use the human data? It seems a swine cohort and a human cohort are simulated virtually, why did the authors do this? More information here would help.
6. Figure 3: I'm not sure I understand how this is a validation of the model fit. Usually, a validation compares the model output to some new data but I couldn't see any data in Figure 3. Could the authors elaborate further on this. Could the authors also explain how many simulations were used to calculate the STD. It says 2 in the caption, but I'm assuming it was more than 2, as 2 is not sufficient to represent the STD of a model prediction for validation.
7. Supplemental Fig 3: are figures F-J missing? I could not see these plots on the PDF version I have
8. Figure 4: "Model validation in huamns" the caption of this figure suggests the use of human data to fit the model, but there is no data presented in the figure, unless I am misunderstanding. I thought the figure it plotting the ODE model prediction? Can the data used to fit the model be added to the figure?

Reviewers' comments:

Reviewer #1 (Remarks to the Author):

As someone who is very familiar with the good work previously produced by several of the authors, it was a pleasure to review this manuscript. The point of this paper is to develop a computational model that can predict and guide trauma patient outcomes and treatment from limited pre-hospital patient data when supplemented with more detailed animal-model data, collected in the immediate aftermath of severe injury. The work is soundly motivated, timely, and novel. This reviewer can appreciate the scope and complexity of the described model development, given the large number of associated parameters and the overall model size.

I have a number of suggestions to improve the overall clarity of presentation. At a high-level, this boils down to providing more technical detail at certain manuscript locations, as indicated below. Also, the validation process as described is a little unclear, at least in terms of visual comparisons with the actual data, and also in not fixing all of the parameters during the validation stage, which is what should be done.

Thank you for your very generous comments and for your thorough review of our manuscript. We appreciate your insights and your great suggestions and have endeavored to respond to each of them in a way that will truly strengthen our submission.

Line-by-line comments:

Line 74: You may want to point out that evidence for an underlying rationale in support of this hypothesis was recently put forward, and it has to do with how a patient's protein concentrations normalize or not depending on what exactly is provided in the plasma. It seems that patients will survive if, and only if, key protein concentrations normalize. See Ghetmiri et al, npj Systems Biology, 2021.

This is a really fascinating observation. We have revised this sentence accordingly and have added this reference to the manuscript (Line 74):

It is hypothesized that early plasma resuscitation may modulate inflammatory and endothelial cell responses to injury¹⁶ and recent work has associated survival with the necessity to normalize key coagulation protein concentrations [Ghetmiri DE, Cohen MJ, Menezes AA. NPJ Syst Biol Appl. 2021 Dec 7;7(1):44]

Line 86: The two references are for sepsis (and by the same authors). A more apt, albeit older, mechanistic model of coagulation with massive dimensionality is Luan et al, Molecular BioSystems, 2010.

Thank you for this suggestion. We have added this excellent reference.

Line 101: I feel that some description about virtual populations are in order here, before going any further. How were the virtual populations generated? How representative are they, and what are the guarantees of the population statistics? For what reason was the virtual population generated: testing or updating the model? How do the generated population statistics (differing from the actual statistics) affect the goal of testing or model updates (i.e., what are the biases, if any?)

These are great questions, and we agree that some context should be provided to explain these virtual populations at this point in the manuscript. The virtual populations were generated by uniformly sampling pre-liver injury parameters from values found in fitting to generate virtual populations for different theoretical arms/protocols. We have revised this sentence to provide some of these additional details at this early phase of the manuscript as follows (Line 102):

To generate virtual animal populations, we uniformly sampled parameters from values found in fitting and then tested variations in the trauma resuscitation in silico (Fig 2). For the human experimental protocols, n=10 patients from the fitting cohort (all 5 non-survivors who received at least 1 plasma infusion and 5 matched survivors) were fit to individual data on a 1:1 basis (Fig 2).

Line 123: "local initial conditions were allowed to vary over the span of ranges observed in the fitting data set" This is unclear in light of the previous sentence on locking (globally) vs the non-locking locally (next sentence), and also the emphatic distinctions you are drawing on verification (really, just continued model fitting) compared to validation. For this line 123, why were local initial conditions allowed to vary? At the end of the day, since this is holdout data, it makes more sense to check real vs. predicted outputs for fixed initial conditions, especially since n=4 (small number). Varying initial conditions will impact predictions and affect accuracy.

This is an important methodologic point that should be clarified. This approach allowed the model to maintain an initial healthy steady state with non-zero initial pro-inflammatory cytokine levels that reflected initial animal interventions. We have modified the text as follows (Line 128):

To maintain an initial healthy steady state with non-zero initial pro-inflammatory cytokine levels, local initial conditions were allowed to vary...

Line 125: "fit targets accurately" How accurately? What are the metrics? There are large ranges, so how good is this fit overall?

The goodness of fit measures are on an arbitrary scale, and so they do not have any meaning other than the ability to monitor the fit from run to run to track improvements to the score. In the model fitting process, we noted a baseline error

function score, and the scores were in this same range during the verification and validation phases of this study.

Line 126: Fig 3. Legends and figure panels should explicitly show and state experiment and predicted.

Thank you for this comment. We have added the actual experimental data to each panel in this Figure and have indicated that the lines and shaded areas are predicted model fits in this verification stage.

Line 131: Fig 4. First, it is unclear what is being plotted, predictions or data. (I suspect just predictions, but this is not completely clear from the caption or the panel legends.) For validation, as is stated as the goal in the caption, both predictions and data should be shown for visual comparison. (I acknowledge that showing all data can be messy, but it is useful.) Because, if these are just predictions, then how do the lines show validation against actual? And if they are data (e.g., the shaded regions) vs. the model (e.g., the lines), then the caption doesn't make sense.

Thank you for this comment. We have added the actual human data to the panels where human data exist (blood pressure, C and O2Sat, E). The “Bled Volume, “EPAL” and all “Damage” parameters are all derived, so there is no physiologic measure to include. We have also updated the caption accordingly.

Line 138: "accurately categorized." How accurate? What are the metrics?

The model perfectly categorized all survivors (n=10) and non-survivors (n=6) in the validation cohort, and for the non-survivors, it predicted the time of death to within 1 minute of the actual time of death. We revised the end of this paragraph to articulate these points more clearly, as follows (Line 144):

Despite this, the calibrated model accurately categorized all survivors (n=10) vs. non-survivors (n=6) in the validation cohort based on differences in the damage (DAMAGE), trauma-specific damage (TRAUMA_DAMAGE), and cumulative damage (AUC_DAMAGE) variables (Fig 4, Table 1) and for the non-survivors, the model predicted the time of death to within 1 minute (0.67 ± 0.82 minute).

Line 142: This section is about linking porcine and human data. But after reading this section carefully, it wasn't clear to me what the linkage is. The first two of the three paragraphs in the section are about swine, the last paragraph is about human. Where is the paragraph about the linkage? For example, is the connection that some/all parameters in the swine-tuned model were scaled-up for, or turn out to be a scaled-up version of, the human-tuned model? Which parameters? Or is the connection that some output from the swine-tuned model fed into the

human model? If so, which ones, why and how? Essentially, the linkage needs to be clearly articulated. This gets to the "integrate" in the title of the paper.

This is an essential point—thank you for requesting this clarification. We have revised the introductory paragraph of this section to more clearly articulate how the models were integrated/linked (Line 149).

The final model that was re-parameterized to best reflect both porcine and human data was fit within 9.8% of the original goodness of fit score..... The final re-parameterized model was then used to conduct exploratory in silico studies in both animals (Fig. 2C) and humans (Fig. 2D) to assess the performance of plasma resuscitation timing and volume.

In addition, we have placed the initial conditions and the parameters for both the animal and human models side-by-side in our supplemental materials for easy reference.

Lines 147-149: How were the virtual animal populations developed, how different are they, how do they examine the effect of therapies? You did seven different populations for each treatment, why not use one population and check each of the seven treatments on the same population?

The reviewer brings up a very interesting point that is not discussed often when considering how to structure in silico studies such as those described in our manuscript. There certainly is merit to the idea that simulations allow for assessment of a given perturbation in a population much larger than that likely to be used in real-world studies. However, this *in silico* scale-up is also a potential concern because typical animal experiments only utilize numbers in the 10s, and typical observational clinical studies (not randomized clinical trials) utilize, at most, numbers in the 100s. It therefore is possible that conclusions from simulations of much larger populations may lead to conclusions that are correct yet not verifiable in real-world contexts. In an attempt to avoid this type of problem and yet still sample the vast state space of these simulations, we carried out multiple different population studies with numbers of simulated subjects as indicated. We have added this clarification to our methods section (Line 470):

All experimental populations were generated to have 100 unique members to sample a significant portion of the state space with these simulations using a number of animals in which our findings could reasonably be verified.

Lines 154-156: Is it three or is it five human populations, based on Figure 2D and 5? Typo?

We apologize for any error or lack of clarity. There were five human *in silico* experiments based on giving plasma pre-hospital (Exp 1), starting the resuscitation in

the hospital sooner by either 30 minutes or 10 minutes (Exp 2 and 3, respectively), or delaying resuscitation in the hospital by 10 or 30 minutes (Exp 4 and 5, respectively). We modified this section to hopefully make this more clear and also added a reference to Fig 2D to provide a visual reference for the five experiments (Line 163):

Subsequently, five human in silico experiments assessed giving plasma pre-hospital (Exp 1), starting the resuscitation in the hospital sooner by either 30 minutes or 10 minutes (Exp 2 and 3, respectively), or delaying resuscitation in the hospital by 10 or 30 minutes (Exp 4 and 5, respectively) (Fig. 2D, 5).

Lines 156-159: Simulation and original trajectories are not shown in the figure, so how to compare?

The original, baseline trajectory for each parameter is shown in Fig 5A and then the results of each experiment are shown in Fig 5B to F.

Line 160: It seems to me like the goal should be made really explicit. Are you planning to evaluate how much the porcine data negatively impacted the human prediction, since the human data is the same for training and predictions? It would be useful to see predictions with and without the porcine learning, to assess overall impact of the porcine data. This goes to the significance of the study.

While we did not consider evaluating the negative impact of porcine data on the human prediction, we agree that this is a valuable suggestion. The approach suggested by the reviewer could be useful to better quantify the value of the initial step we took in building our model around the more granular animal data to then import the overall framework of the human model that then used much more sparse data for training, verification, and validation. This analysis would exceed the scope of our current work, but we look forward to exploring this concept going forward. Thank you again for this very good comment. We have added the following to our limitations section (Line 316):

Finally, we did not assess the effect of human modeling results with and without the benefit of initial model development and training/verification on animals.

Line 169: "over baseline." What is the baseline? Is it per patient, or is it the mean in the population?

We apologize for any lack of clarity. The baseline referenced is the calculated, per-patient bled volume for the 10 patients used in the *in silico* experiments. This has been clarified in the text (Line 180):

This experiment suggests that giving the first plasma infusion earlier (15 minutes post-injury) increased bleeding by a $5.2\% \pm 1.2\%$ (mean \pm standard deviation) over baseline bled volumes on a per-patient basis,...

Line 169: "5.2% plus/minus 1.2%" Somewhere near here, it needs to be confirmed that this is mean plus/minus 1 std dev?

Thank you for this comment. The standard deviation was the actual standard deviation of the mean in the data, not mean \pm 1 or more standard deviations. We added a mean \pm standard deviation label here for clarification as per our revision above.

Lines 169-171: The percent results are small. So how useful are these numbers? A remark needs to be added. Same for line 180, 184.

We agree, these numbers are small. As such, they suggest a benefit that warrants further exploration, and that will likely only appear in large populations. This further reinforces the concept that no one simple improvement will broadly result in improved outcomes for all patients. Indeed, recent clinical data indicate that early plasma resuscitation has the greatest impact on polytrauma patients with TBI, supporting the concept that specific resuscitation regimens largely benefit specific patient sub-populations. We have added additional commentary in the locations you suggested. For example, Line 183:

In a clinical setting, the additional bleeding would be expected with the improved blood pressure that resulted in our modeling, and although the improved survival time and decreased damage were modest, these incremental benefits are consistent with recent clinical studies indicating early plasma has the greatest benefit for combat casualties [Shackelford et al JAMA] and polytrauma patients with TBI [Sperry et al, PAMPER, NEJM 2018; Gruen et al, JAMA Netw Open 2020]. Our findings also suggest that a single intervention alone is unlikely to result in widespread improved clinical outcomes for all patients at risk, yet this approach can allow us to refine the optimal target patients for each intervention while also assessing the effect of combination therapies, all warranting further exploration.

Lines 370-371: It is not clear what ODEs are used, or what type of solver was run to calculate the ODEs. The ODEs are not specified separately in the supplement as is typical for these sorts of papers (although it appears they are embedded in the code, but these equations are hard to read and parse). Initial conditions and parameters are made very clear. Having a description of the ODEs being solved in each compartment in the supplemental materials would be helpful.

Thank you for these comments; we apologize for this omission. We have now written out all of the ODEs previously provided only as code. These are now provided as

supplemental files. The solver used was a custom platform generated by Immunetrics, but we note that it is largely compatible with XPP.

Lines 409-412: There is no mention of tissue factor even though this is a key part of the extrinsic signaling cascade associated with trauma. Is tissue factor considered at all in the model, and if not, why was it chosen to be left out?

This is an excellent question. Qualitatively speaking, tissue factor (TF) is intrinsic to the “Coagulation” oval in Fig 1 and the “Active Procoag” rectangle in Suppl Fig 1. But, as you point out, we did not have specific TF assay values to inform our model quantitatively as this is not generally available due to very low circulating levels even after injury. Nonetheless, we did measure downstream effects with VII activity (we fixed a typo that listed factor VI rather than VII in two locations). We have added an acknowledgment that our reductionist approximation of the coagulation system does not come close to capturing the vast complexity of the clotting cascade or the cellular theory of coagulation in a new sentence in our limitations section (Line 292):

Thus, any attempt to model these incompletely understood and highly variable systems will necessarily represent an abstraction that, at best, approximates reality. As an example, the animal phase of model development assumed that subjects were relatively hyper-coagulable based on the literature⁵³. However, the degree of hyper-coagulability and the necessary adjustments to the model for use in humans were both unknown, potentially compounding inaccuracies due to partial quantification of the various elements of the coagulation system and the lumping of all coagulation factor levels into an aggregate measure.

Line 410-412: Summing/averaging the coagulation factor data together is somewhat confusing. The factors are often measured in different units, and have a wide range of concentrations so often one factor is present at a much larger magnitude than another factor. Therefore, if these numbers are averaged and used as general pro-coagulant and anti-coagulant single measures then they might not account for smaller concentration changes occurring in individual factors, which may have a large impact on the overall cascade.

This is also an excellent point. This is clearly an opportunity for us to improve our modeling in future iterations to capture these subtle nuances in the effect of the various coagulation factors. Based on this comment and your comment above, this has to be acknowledged as an additional limitation, which we have done as per our response above.

Lines 502-512: So if local parameters are being allowed to vary, while others are locked, then this is **not** validation (as in the subsection title). Earlier, you call this verification. In general, the whole manuscript needs to be really clear about what is validation: everything should be fixed, and you compare predictions against data that was not used for learning.

We thank the reviewer for pointing out this area of unclarity, for which we apologize. To make our methods section clearer, we revised the headings for both the “Porcine T/HS simulations” (line 377) and “Human trauma simulations” (line 467) to “Porcine T/HS modeling” and “Human trauma modeling” to set the stage for each section.

Then, we have revised the section prompting your comment above as follows (Line 522):

As in the pig-data fitting, model verification used a half-fit-half-predict approach: the global parameters from the fit stage were locked in verification, but the local parameters were still allowed to vary to fit the data. Again, intervention schedules were known, and local parameters were fit, but global parameters (this time including death and damage) were locked. 17 patients (8 die, 9 survive) - separate from the fitting group - were initially used in verification. There were 16 additional patients (7 die, 9 survive) who were subsequently used for final model validation in a “true hold-out” approach - no changes could be made to the model at this point.

Lines 516,517: auc_damage is written differently than elsewhere in the manuscript.

We have fixed this formatting error for consistency throughout.

Remark on author contributions: for a modeling paper, even if the model is large, this paper seems to have a rather large author listing.

Thank you for this comment. We agree that this is a large number of authors but can confirm that all have made substantive contributions to this work. By way of explanation, this project involved Department of Defense collaborators, civilian trauma groups on both the basic/translational science side (Oregon Health and Science University) and the human subjects side (UT Houston), and modeling team members. Consequently, the author list quickly grew to encompass the number on the masthead of our submission.

Figure 5: same comments as were made for Figure 4. Also, Figure 5B the blood pressure graph scale changes, making it much harder to distinguish a difference between the patient's original trajectories (shown in Figure 5A) and the simulated change of getting plasma much earlier. The y-axis should be adjusted so the scale remains the same across all columns of the figure to make comparisons easier. Same comment for 5D.

This is an important observation. *We have added blood pressure data points to 5A and have fixed the scale issues with 5B and D.*****

Supp Figure 2: why do some graphs have marker points but other graphs only have lines? What do these points represent? Additionally, what are the units on the y axis of each graph?

The markers represent experimental data whereas the lines represent mean values from the model. We have updated this figure and the legend to make it consistent with Figure 3 in the manuscript and to clarify these points. We have also added units to all figures throughout, where applicable.

Supp Figure 3 is missing the bottom half of the figure (parts F – J)

Our apologies for this. We will ensure this figure converts properly in the editorial system when we upload our revisions.

Remark on Title: The concept of a digital twin is still relatively novel and there is no universal definition just yet. However, it is generally accepted that a digital twin is more than a personalized model, since it allows for some sort of patient feedback (either in real-time, or delayed through measurements and periodic model inputs), to output updated patient trajectories. Since this model has no real-time patient feedback component, I would argue that it is more of a personalized model than an actual digital twin.

You make a great point. We feel the generation of personalized models is a significant step at the very least TOWARD creating digital twins for hemorrhagic shock but also acknowledge and agree with your point about the lack of real-time patient feedback. We have revised our title as follows and hope this will be acceptable:

Toward Hemorrhagic Shock Digital Twins: Personalized Mathematical Models Integrate Large-Animal and Human Inflammation, Coagulation, and Resuscitation Data

Reviewer #2 (Remarks to the Author):

I like the idea behind this effort, which was to "unify pre-clinical and clinical studies in order to realize the full promise of mechanistic computational modeling, which includes the potential for patient-specific simulations (i.e. "digital twins") and simulated populations (i.e. in silico clinical trials)". This is, however, not a novel concept/approach. Similarly, their conclusions that "resuscitation with plasma and red blood cells together outperformed resuscitation with crystalloid or plasma alone, and that earlier plasma resuscitation reduced both morbidity and mortality" are not novel.

So, while I like the academic exercise, I don't see how this paper adds much to the existing literature. At best these integration efforts provide a rough approximation of the reality, that fails to capture the person-to-person variability. This is a major reason why interventions that work in well controlled animal studies fail (or are not as effective) in human trials.

We thank the reviewer for taking time to review our manuscript. While we understand and appreciate the perspective, we maintain that our approach is novel since it utilizes mechanistic mathematical modeling to connect preclinical to clinical data, and because it reports on the first multicompartment, mechanistic mathematical model that integrates coagulation with inflammation and, ultimately, an abstracted view of whole-patient (patho)physiology (i.e., the “damage” variable). Furthermore, metrics derived from the “damage” variable capture both the acute injury burden and the cumulative effects of the innate responses and treatments, which comprises another novel contribution that can be generalized to other disease states.

The integrated modeling platform described in this manuscript illustrates the complex and dynamic nature of the interactions among the various physiologic systems involved in the response to trauma and hemorrhage. This manuscript also highlights the difficulty inherent in extrapolating from preclinical studies to the ultimate outcomes of clinical trials due to the highly individual nature of the response to severe hemorrhagic shock. We maintain that this platform will allow for *in silico* clinical trials after tuning the models with a small amount of animal and human data (pilot study size datasets) and will thereby allow us to more appropriately focus on potentially viable treatment strategies.

Reviewer #3 (Remarks to the Author):

Brief summary of the manuscript

Hemorrhagic shock, the depletion of intravascular volume through blood loss to the point of being unable to match the tissues demand for oxygen, is investigated using a digital twin set-up based on animal/human data. Linking preclinical data in swine and clinical data from patients via a 3- compartment ODE model of inflammation and coagulation, the authors seek to identify intervention mechanisms. Their model predicts accurately measurements, such as time of death in patients, and is used to conclude that plasma and RBC infusion results in increased clotting and reduced bleeding.

Overall impression of the work

The use of mathematics and digital twins is an exciting area of research and it is fantastic to see an application into Hemorrhagic shock. Unfortunately, I find the details surrounding the mathematics insufficient to reproduce the work and to critically analyse the validity of the results presented by the authors. If the authors could provide more details about their model, and the methodologies they used, this would significantly support their manuscript and make this a very impactful piece of work. I would be happy to review a revised version of their manuscript.

Thank you for your very gracious comments. We have thoroughly revised our manuscript in response to your comments and those of the other reviewers and have also provided the model ODEs in both code and mathematical formats for your consideration. We sincerely hope these additions and revisions will permit further critical analysis of the validity of our work.

Specific comments, with recommendations for addressing each comment

1. Mathematical modelling details: “Model parameters, initial conditions, and code files are provided in Supplementary Materials”, in my opinion the details provided in the supplementary materials are insufficient in their current form to accurately describe the model so that it can be (i) critically understood and (ii) reproduced. Could the authors please provide the list of ordinary differential equations, along with the meaning of the terms in the equation, motivation (from the literature or otherwise) for the base forms of mechanisms being modelled, details of mechanisms that were omitted and then how each of the parameters were estimated. In the Materials and Methods a lot of the assumptions are listed in terms of their biology but there is no reference to the modelling terms that represent them.

We have converted our code into the mathematical equations they represent for more straightforward viewing. We have more clearly defined the meaning of each parameter and have included references for the base form of the equations, as appropriate. In developing the coagulation module, we discussed multiple iterations before settling on the current form. We have provided revisions to the manuscript to capture some of the inherent limitations in our reductionist approach in response to Reviewer 1 (p5 above) and in response to your comments. We hope these additions to our supplemental content and our revisions provide sufficient detail.

2. 3 compartment ODE: Further to my above comment, without reading Brown et al. it's not clear how Figure 1 depicts a 3 compartment ODE model. Usually, this phrase means there are only 3 ODEs, however, I think based on the simulations there are more? What do all the nodes and arrows represent in their figure in each of the ODEs? More details on the equations, exactly what each variable means etc is needed. Minor comment: caption should be able to summarise the figure without needing to read the text, more detail on what different coloured arrows mean, different coloured boxes etc is necessary.

Thank you for these comments. We apologize for any confusion in our description of the model. Figure 1 does not depict a 3 ODE model but rather an abstracted version of a pig/human consisting of multiple body compartments within which extensive physiology and inflammation biology is represented by multiple ODEs.

We have now added a supplement that includes all model equations comprising the simulated body compartments (tissue, blood, etc.) and a simplified version of the model (Supplemental Figure 1) to better explain and detail the abstracted version of the model depicted in Figure 1. We have also added additional details to the caption

of Figure 1 to better explain the figure along with a new supplemental figure to distill the effects of these various factors on the critical “damage” metric (new Supplemental Figure 1). Finally, we have provided a narrative explanation of the model as a supplemental file (Model Narrative).

3. Fitting algorithm: There is insufficient information on how the parameters were fit in the model and which parameters were fit. For example, what fitting algorithm was used? How was the uncertainty in the fitting evaluated? What confidence intervals were obtained? How many parameters were fit to how many data points. Also where are the images for the model fit to data?

The model parameters were fit by minimizing a weighted least squares objective using a sequential Monte Carlo method. Confidence intervals were not obtained (note that these are individual patient fits). 33 parameters were fit. There were four measured analytes in the data: O2Sat, blood pressure, platelets, time of death; where patients had different numbers of points for some analytes (predominantly blood pressure). There are also heuristics constraining IL1, IL6, IL10, TNF, blood pressure, nitric oxide, and bleeding. We have revised our figures/supplemental figures to make these details more clear and have added verbiage to our methods (Line 447):

A total of 33 parameters were fit by minimizing a weighted least squares objective using a sequential Monte Carlo method.

4. Virtual individuals: How were the number of virtual individuals in each group chosen? E.g. why were there only 10 in each group for the humans? The authors say “we uniformly sampled from the range of values found for locally fit parameters to generate virtual populations for different theoretical arms” was this only for the swine virtual trial? Or also for the humans? but then why were only 10 humans in each group used? How did the authors confirm their virtual population was representative of a sample from their interval? Why did the authors not sample more human patients from an interval returned for the fitted data?

For the human portion of this study, we chose 5 survivors and 5 non-survivors, all of whom received at least one plasma infusion, to run the theoretical human experiments. These individuals were fit to individual data on a 1:1 basis. The sentence “...we uniformly sampled from the range of values...” only applies to the swine virtual trial.

Hemorrhagic shock with coagulopathy carries an exceedingly high mortality rate, and death occurs within the first four to six hours of presentation to the hospital. Our virtual human populations (5 survivors, 5 non-survivors) reflect the upper limit of mortality in this population. In our training dataset of n=35 patients, only 5 non-survivors received a unit of plasma; so we used all of these patients and 5 matched survivors. We have revised our introduction and methods to provide this justification (Line 53; Line 166):

In these patients, death occurs early (within 1 hour) [Oyeniya BT et al. Injury. 2017 Jan;48(1):5-12], and for early survivors hemorrhage and trauma induce an acute inflammatory response that ultimately drive multiple organ dysfunction and death much like sepsis.

Each in silico experiment was carried out using 10 of the patients from the calibration set (using all 5 non-survivors who received at least 1 plasma infusion to reflect the high mortality in this population and 5 matched survivors).

5. Motivation for swine and human in silico trials: I'm not sure I understand why both the pig and human data were used? Why not just use the human data? It seems a swine cohort and a human cohort are simulated virtually, why did the authors do this? More information here would help.

As the reviewer is likely aware, is usually difficult to carry out multiple experimental manipulations or get extensive time course data from humans, while these are easier to perform in animals. In contrast, key outcomes such as mortality typically occur artificially in experimental animals (i.e., animals are euthanized at the end of an experiment, and, for animal welfare reasons, usually cannot be left to succumb to hemorrhagic shock). In contrast, mortality is a key natural outcome in severely injured humans. This is why drug and device development require experimental manipulation in large animals and validation in humans.

In our study, the swine data was collected under reasonably controlled conditions and was able to provide some degree of "control" for the insults experienced by human patients. By training the model to this, we were able to gain more confidence in the model behavior before moving onto individual human fits. We have noted these motivations in the Introduction (Line 86).

In the present study, we sought to bridge physiologic, inflammatory, and clinical aspects of T/HS in large-animal pre-clinical studies (in which early responses to experimental perturbations could be assessed *in very granular detail under controlled conditions* but outcomes are limited to several hours after the insult) with human clinical studies (in which early data are sparse and inter-individual variability is high).

6. Figure 3: I'm not sure I understand how this is a validation of the model fit. Usually, a validation compares the model output to some new data but I couldn't see any data in Figure 3. Could the authors elaborate further on this. Could the authors also explain how many simulations were used to calculate the STD. It says 2 in the caption, but I'm assuming it was more than 2, as 2 is not sufficient to represent the STD of a model prediction for validation.

We apologize for the lack of clarity and have added symbols depicting the experimental data to the appropriate panels. There were 2 animals in each group resulting in a mean value for the fitted mean represented by the line and a standard error of the fitted mean represented by the shaded area.

7. Supplemental Fig 3: are figures F-J missing? I could not see these plots on the PDF version I have

Our apologies for this. We will ensure this figure converts properly in the editorial system when we upload our revisions.

8. Figure 4: "Model validation in huamns" the caption of this figure suggests the use of human data to fit the model, but there is no data presented in the figure, unless I am misunderstanding. I thought the figure it plotting the ODE model prediction? Can the data used to fit the model be added to the figure?

We have added human subject data to panels C (blood pressure) and E (O2Sat) for improved clarity.

We sincerely thank the editors and all the reviewers for their time in reviewing our work and for providing these thoughtful comments.

Reviewers' comments:

Reviewer #1 (Remarks to the Author):

Thank you very much for thoroughly revising your manuscript in response to my prior concerns, and for explaining your edits. I only have a few minor comments, as indicated below.

Line-by-line comments:

Line 102: The last part of this paragraph is still a little unclear, and confusing when reading the next subsection. I suggest adding something similar to the following sentence at Line 102, right before the sentence that starts "To generate virtual...."

Sentence to add should go something like: "Model outputs informed tests of simulated therapy strategies on virtual animal populations and a subset of the human patient data that the model was developed from."

Line 105: suggest the following to clearly describe the test:

"human experimental protocols"  "human simulated experimental protocols"

Lines 330-331: As the authors are already aware, there is a community effort to move away from using animal models altogether, since the practice is considered barbaric, brutal, and completely unnecessary because the results are not sufficiently informative to justify torture, and the practice is dismissive of the innate value of life while adding untold suffering. This is no matter any IACUC protocols that may or may not have been followed. Accordingly, I take issue with the wording in these two lines, since the wording condones and even recommends further animal cruelty.

A small suggested change at Line 330 will have a profound impact:

"assessed with a small number"  "assessed with existing data from a small number"

Line 366: "define damage as a function of ..."

I tried to locate this function (mostly because I was interested in seeing if it was linear or not, and what the weights were), but was unable to find this function in the Supplement. Could the authors please provide this function explicitly, and also provide the equation number reference at this location if the function is in the Supplement.

Line 536: This sentence is abrupt, and needs to be edited.

Line 540: As in Line 105, add word "Simulated" before "Experimental"

Figure 2D: has a spelling error for Infusion

Supp: High Level Overview of the Model. Related to Line 366 comment, I was unable to find the value of the threshold triggering the state of death earlier in the manuscript or at this location. Both the damage function and the threshold for AUC_damage to result in death should be articulated somewhere, and if they already are, referred to with equation numbers.

General comment: Integer numbers less than ten should be spelled out in the manuscript text, as long as the number is not part of a name (e.g., Experiment 2 is fine; but it should be three-compartment, five survivors, within six hours of admission, seven arms, and so on).

Line 269: *in silico* should be italicized.

Reviewer #2 (Remarks to the Author):

The authors have done a solid job addressing the concerns. I have no additional questions

Reviewer #3 (Remarks to the Author):

I appreciate the revisions the authors have made to their manuscript.

I think for transparency for the readership, more should be said about the potential limitations of the parameter fit.

The authors mention adding the following to the manuscript: "A total of 33 parameters were fit by minimising a weighted least squares objective using a Monte Carlo method". But nothing else. This is a considerable number of parameters, so the degrees of freedom should be given (i.e. the number of data points being fit to-1) so that readers know how much uncertainty there may be in this estimation.

In response to my question around confidence intervals the authors said "note that these were individual patient fits". I'm not quite sure why that would motivate not obtaining confidence intervals and as confidence intervals can be obtained from least squares objective minimisation I'm not sure why the authors wouldn't have obtained them. Either way, a comment regarding the uncertainty in the fit needs to be included given the highly translational nature of this work. Without obtaining the confidence intervals, how can the authors be sure their parameter estimates are the correct values.

Apart from this minor concern, I'm very happy with the revised manuscript.

Reviewers' comments:

Reviewer #1 (Remarks to the Author):

Thank you very much for thoroughly revising your manuscript in response to my prior concerns, and for explaining your edits. I only have a few minor comments, as indicated below.

We very much appreciate your thoughtful review and insightful comments. We have responded to your further comments and requests below, and hope that, in its further revised form, our manuscript will be acceptable for publication.

Line-by-line comments:

Line 102: The last part of this paragraph is still a little unclear, and confusing when reading the next subsection. I suggest adding something similar to the following sentence at Line 102, right before the sentence that starts "To generate virtual...."

Sentence to add should go something like: "Model outputs informed tests of simulated therapy strategies on virtual animal populations and a subset of the human patient data that the model was developed from."

Thank you for this comment. We have added a sentence at line 102 as you have suggested to further clarify our approach and intent.

Line 105: suggest the following to clearly describe the test:

"human experimental protocols"  "human simulated experimental protocols"

We have added "simulated" as suggested.

Lines 330-331: As the authors are already aware, there is a community effort to move away from using animal models altogether, since the practice is considered barbaric, brutal, and completely unnecessary because the results are not sufficiently informative to justify torture, and the practice is dismissive of the innate value of life while adding untold suffering. This is no matter any IACUC protocols that may or may not have been followed. Accordingly, I take issue with the wording in these two lines, since the wording condones and even recommends further animal cruelty.

A small suggested change at Line 330 will have a profound impact:

"assessed with a small number"  "assessed with existing data from a small number"

We agree and have made this revision.

Line 366: "define damage as a function of ..."

I tried to locate this function (mostly because I was interested in seeing if it was linear or not, and what the weights were), but was unable to find this function in the Supplement. Could the authors please provide this function explicitly, and also provide the equation number reference at this location if the function is in the Supplement.

This is a great point. The functions for damage are not ODEs so we did not include them in our supplemental equations. We have now added these specific equations to the body of the manuscript at this point. An earlier version of our manuscript included the model code, which contained these equations and which we have now added back as supplemental content.

$$\text{bp_damage} = k_damage_bp * \text{fm}(\max(\text{bp_damage_threshold} - \text{blood_pressure}, 0), x_damage_bp, 2)$$

$$\text{O2Sat_damage} = k_damage_O2Sat * \max(\text{O2Sat_damage_threshold} - \text{O2Sat}, 0) / \max_O2sat$$

$$\text{il6_damage} = k_damage_il6 * \text{fm}(\max(\text{il6} - \text{IL6_damage_threshold}, 0), x_damage_il6, 2)$$

$$\text{trauma_damage} = k_damage_trauma * \text{trauma} / \max_ISS$$

$$\text{Damage} = \text{bp_damage} + \text{il6_damage} + \text{O2Sat_damage} + \text{trauma_damage}$$

$$\text{where } \text{fm}(v, x, \text{Hill}) = \frac{v^{\text{Hill}}}{v^{\text{Hill}} + x^{\text{Hill}}}, \max(\text{O}_2\text{Sat}) = 98, \text{ and } \max(\text{ISS}) = 75.$$

Line 536: This sentence is abrupt, and needs to be edited.

This section has been edited to read more smoothly as follows: A global death threshold is set in fitting, and when AUC_DAMAGE exceeds this threshold, death is triggered. However, even if death is triggered, the model simulation continues out to six hours although the trajectory after death is triggered remains moot.

Line 540: As in Line 105, add word "Simulated" before "Experimental"

We thank the reviewer for this suggestion, which we have implemented.

Figure 2D: has a spelling error for Infusion

This has been fixed.

Supp: High Level Overview of the Model. Related to Line 366 comment, I was unable to find the value of the threshold triggering the state of death earlier in the manuscript or at this location.

Both the damage function and the threshold for AUC_damage to result in death should be articulated somewhere, and if they already are, referred to with equation numbers.

No animals died during the course of the experimental procedure itself; therefore, a threshold could not be set for the animal arm of this study.

For the human arm, damage_death_threshold was tuned and then locked using the PROMMTT data. The threshold is 276.53. This has been added to the manuscript.

General comment: Integer numbers less than ten should be spelled out in the manuscript text, as long as the number is not part of a name (e.g., Experiment 2 is fine; but it should be three-compartment, five survivors, within six hours of admission, seven arms, and so on).

This has been rectified throughout.

Line 269: *in silico* should be italicized.

Done.

Reviewer #2 (Remarks to the Author):

The authors have done a solid job addressing the concerns. I have no additional questions

Thank you for your thoughtful review and for your encouraging comment.

Reviewer #3 (Remarks to the Author):

I appreciate the revisions the authors have made to their manuscript.

I think for transparency for the readership, more should be said about the potential limitations of the parameter fit.

We agree and have added the following limitations related to parameter fit to the discussion: Furthermore, our approach to parameter fitting carries multiple potential limitations, including use of some parameters from literature sources that may or may not apply, intrinsic non-identifiability of the real and model systems with associated uncertainty about the parameter estimates obtained by this process, uncertainty regarding possible outputs of the system, and existence of additional, equally plausible model structures and/or parameter regimes that would fit the data equally well.

The authors mention adding the following to the manuscript: "A total of 33 parameters were fit by minimising a weighted least squares objective using a Monte Carlo method". But nothing else. This is a considerable number of parameters, so the degrees of freedom should be given (i.e. the number of data points being fit to-1) so that readers know how much uncertainty there may be in this estimation.

On average, the number of data points per subject included the following:

Set	Animals	Average number of data points per animal								
		BP	RBC	Plt	Mono	Nu	IL-6	TNF	Anti	Pro
Fitting	12	11	7	7	7	7	3	3	7	7
Verification	4	11	7	7	7	7	3	3	7	7

*Values less than indicates data points were missing in some patients.

Set	Patients	Average number of data points per patient*			
		BP	O2Sat	Plt	t_death
Fitting	35	2.4	0.571	0.914	1
Verification	17	2.941	0.529	0.824	1
Validation	16	2	0.438	0.875	1

*Values less than indicates data points were missing in some patients.

However, degrees of freedom (DOF) in our nonlinear models must be interpreted with caution. Also, some parameters are estimated per-individual while others are estimated universally, further complicating attempts at DOF estimation and interpretation. We moved the Monte Carlo modeling comment to the general model section of Materials and Methods (Line 405) and added comments on model uncertainty as follows: A total of 33 parameters were fit by minimizing a weighted least squares objective using a sequential Monte Carlo method. Factors influencing uncertainty during model fitting included degrees of freedom (range zero to ten for measured analytes), non-linearity of the model, and parameter estimation on an individual basis vs cohort basis.

In response to my question around confidence intervals the authors said "note that these were individual patient fits". I'm not quite sure why that would motivate not obtaining confidence intervals and as confidence intervals can be obtained from least squares objective minimisation I'm not sure why the authors wouldn't have obtained them. Either way, a comment regarding the uncertainty in the fit needs to be included given the highly translational nature of this work. Without obtaining the confidence intervals, how can the authors be sure their parameter estimates are the correct values.

We agree that the estimated parameters for our model, and indeed any such non-linear model, are almost certainly not correct. Indeed, one can debate whether "absolutely correct" parameters even exist. It is also not clear that one-dimensional

views of parameter uncertainty are meaningful in multidimensional, nonlinear models such as ours since many tightly constrained parameters can still interact in complex ways and produce surprising behavior. We therefore believe it is, on balance, better to focus on variability in prediction output. We added comments about parameter fit uncertainty as per our response above.

Apart from this minor concern, I'm very happy with the revised manuscript.

Thank you for your careful review and great comments that have served to improve our manuscript significantly.

REVIEWERS' COMMENTS:

Reviewer #1 (Remarks to the Author):

I thank the authors for addressing my previous comments. I have no substantive concerns, and I recommend that this manuscript be accepted for publication. I applaud the authors on their scholarship, and I look forward to seeing their published article in due course.

A note: In this round of review, six pages of supplement that had been added in the previous round because of reviewer comments appear to be missing in this round. Two of these pages were entitled "High Level Overview of the Model" and, importantly, there were four pages of differential equations. Please restore these six missing pages in the final published version of the supplementary materials.

Reviewer #3 (Remarks to the Author):

I am happy with reviewer changes and approved the current manuscript.

Reviewers' comments:

Reviewer #1 (Remarks to the Author):

I thank the authors for addressing my previous comments. I have no substantive concerns, and I recommend that this manuscript be accepted for publication. I applaud the authors on their scholarship, and I look forward to seeing their published article in due course.

A note: In this round of review, six pages of supplement that had been added in the previous round because of reviewer comments appear to be missing in this round. Two of these pages were entitled "High Level Overview of the Model" and, importantly, there were four pages of differential equations. Please restore these six missing pages in the final published version of the supplementary materials.

Thank you again for your thoughtful review and for your kind remarks. We have replaced the "High Level Overview of the Model" and the ODEs in the Supplementary Materials.

Reviewer #3 (Remarks to the Author):

I am happy with reviewer changes and approved the current manuscript.

Thank you